# Structure of the planar cell polarity cadherins Fat4 and Dachsous1

Elliot Medina [1,2], Yathreb Easa[3], Daniel K. Lester[2,4,5], Eric K. Lau[4,5], David Sprinzak [3] & Vincent C. Luca [1,5] ✉

The atypical cadherins Fat and Dachsous are key regulators of cell growth and animal development. In contrast to classical cadherins, which form homophilic interactions to segregate cells, Fat and Dachsous cadherins form heterophilic interactions to induce cell polarity within tissues. Here, we determine the co-crystal structure of the human homologs Fat4 and Dachsous1 (Dchs1) to establish the molecular basis for Fat-Dachsous interactions. The binding domains of Fat4 and Dchs1 form an extended interface along extracellular cadherin (EC) domains 1-4 of each protein. Biophysical measurements indicate that Fat4-Dchs1 affinity is among the highest reported for cadherin superfamily members, which is attributed to an extensive network of salt bridges not present in structurally similar protocadherin homodimers. Furthermore, modeling suggests that unusual extracellular phosphorylation modifications directly modulate Fat-Dachsous binding by introducing charged contacts across the interface. Collectively, our analyses reveal how the molecular architecture of Fat4-Dchs1 enables them to form long-range, high-affinity interactions to maintain planar cell polarity.

The cadherin superfamily is comprised of calcium-dependent adhesion molecules that mediate cell-cell interactions. Besides their adhesive functions, cadherin proteins regulate tissue morphogenesis, tissue homeostasis, mechanotransduction, neural circuit wiring, and several other biological processes[1–3]. There are various subgroups within the cadherin superfamily, including type I and II classical cadherins, protocadherins, desmosomal cadherins, seven-pass transmembrane cadherins, and Fat and Dachsous group cadherins. This latter group, Fat and Dachsous cadherins, are distinguished by their giant size (350–550 kDa) and ability to regulate planar cell polarity (PCP). While classical cadherins engage in homotypic interactions to organize populations of cells[4,5], Fat and Dachsous proteins form heterophilic interactions to influence the formation of global or local polarities[6] (Fig. 1A). This function is achieved because Fat and Dachsous proteins, while co-expressed in

the same cell, are asymmetrically located such that trans-interactions between cells propagate PCP.

Fat and Dachsous proteins were originally discovered in *Drosophila* but are evolutionarily conserved throughout metazoan organisms[7]. Genetic knockouts of Fat or Dachsous cause polarity defects in the fly wing, eye, and abdomen. Additionally, Fat regulates tissue growth through modulation of the Hippo pathway[8], and metabolism through a direct interaction between a cleaved intracellular domain (ICD) fragment and Complex I of the OXPHOS chain in mitochondria[9]. In *Drosophila*, there are two Fat proteins (Ft and Ft-like) and one Dachsous protein (Ds). In vertebrates, there are four Fat proteins (Fat1–4) and two Dachsous proteins (Dchs1–2). Fat1–3 are homologs of Ft-like, Fat4 is the homolog of Ft, and Dchs1 is the homolog of Ds[10]. The PCP effects are conserved in Fat4 and Dchs1, but the proliferative effects appear to be tissue- and context-

[1]Department of Drug Discovery, H. Lee Moffitt Cancer Center & Research Institute, Tampa, FL 33602, USA. [2]Cancer Biology Ph.D Program, University of South Florida, Tampa, FL 33602, USA. [3]School of Neurobiology, Biochemistry, and Biophysics, The George S. Wise Faculty of Life Sciences, Tel Aviv University, Tel Aviv 69978, Israel. [4]Department of Tumor Biology, H. Lee Moffitt Cancer Center & Research Institute, Tampa, FL 33602, USA. [5]Molecular Medicine Program, H. Lee Moffitt Cancer Center & Research Institute, Tampa, FL 33602, USA. ✉e-mail: vince.luca@moffitt.org

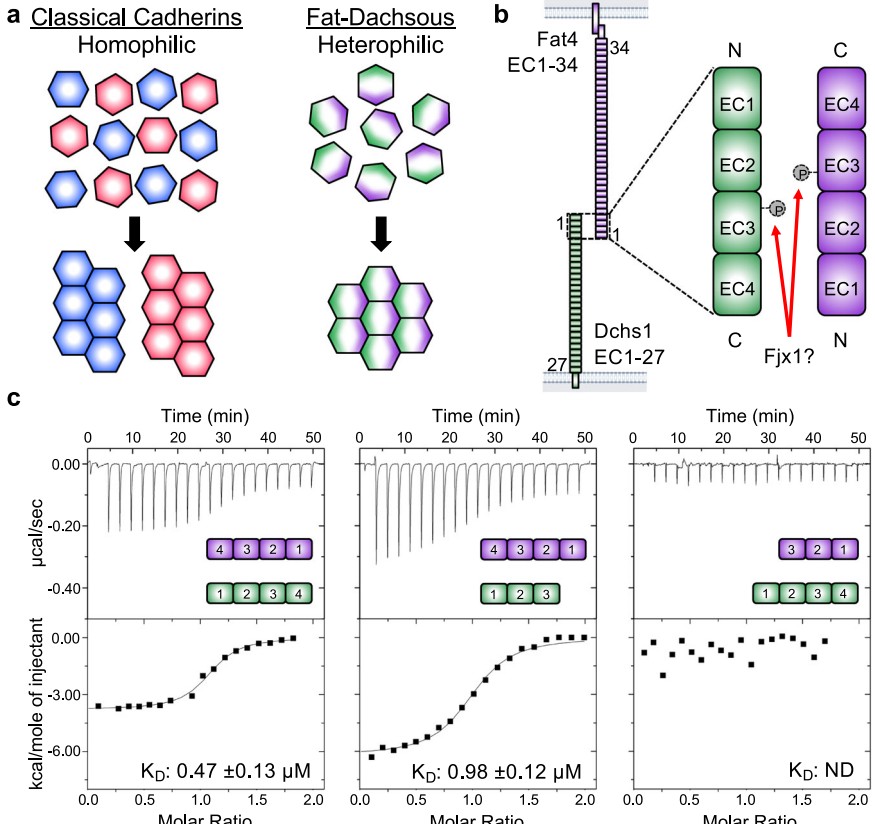

**Fig. 1 | Binding affinity between Fat4 and Dchs1 truncations. a** Simplified cartoon comparing tissue organization by classical cadherins and Fat–Dachsous cadherins. Classical cadherins form homophilic interactions between cells to segregate tissues. Fat and Dachsous are expressed in the same cell and form heterophilic interactions to regulate planar cell polarity. **b** Schematic depicting the Fat4 and Dchs1 ECDs. The interacting EC1-4 domains are enlarged and the putative Fjx1 phosphorylation site is indicated in EC3 of each protein. **c** Representative ITC thermograms measuring the binding between Fat4 and Dchs1 constructs. $K_D$ measurements are an average of $n = 2$ independent experiments. Source data are provided as a Source Data file.

dependent. The metabolic function of Fat was likely co-opted by Fat1[11], and Fat4 seems to have transitioned from a direct regulator of the Hippo pathway to adopting a more indirect role[12], with one example involving sequestration of the Hippo effector YAP (Yki in Drosophila) via Angiomotin-like proteins[13].

Fat4 and Dchs1 play important roles in the development of mammalian tissues[14] and the preservation of stem cell progenitor pools[15–17]. Fat4 or Dchs1 knockout mice die shortly after birth, and pups exhibit curved bodies, neural tube broadening, smaller intestines and lungs, and cystic kidneys[18,19]. In humans, loss-of-function mutations in Fat4 and Dchs1 are associated with the congenital disorders Hennekam syndrome and Van Maldergem syndrome, respectively[17,20]. Additionally, inactivating mutations in Dchs1 cause mitral valve prolapse in zebrafish, mice, and humans[21]. Fat4 has also been classified as a tumor suppressor protein, as loss-of-function mutations and gene suppression are associated with increased cancer cell proliferation, invasiveness, and metastasis[22–27].

Fat4 and Dchs1 are two of the largest members of the cadherin superfamily and contain 34 and 27 extracellular cadherin (EC) domains, respectively (Fig. 1B). Most of the Fat4 and Dchs1 EC domains are connected by calcium-binding motifs that coordinate up to three $Ca^{2+}$ ions[28]. Fat proteins also contain juxtamembrane EGF and Laminin-G domains bridging its EC and transmembrane domains. Despite their large numbers of EC repeats, the N-terminal EC1–4 domains are apparently sufficient for Fat4:Dchs1 interactions[29]. Linear estimates of the size of the Fat4 and Dchs1 ectodomains indicate that the proteins could extend up to 300 nm in length. In microscopy studies, Fat4:Dchs1 interactions have been shown to occur across intercellular spaces ranging from 47 to

116 nm, suggesting a substantial degree of flexibility in their ectodomains[29,30]. There are currently no available structures of Fat4 and Dchs1 alone or in complex, and besides structures of Protocadherin 15 (PDCH15):Cadherin 23 (CDH23) complexes, we generally lack structural descriptions of heterophilic cadherin interactions[31].

The Golgi-associated kinase, Four-Jointed (Fj), regulates signaling between Drosophila Fat and Dachsous by phosphorylating the EC domains of each protein[32]. One current model suggests that Fj-mediated phosphorylation impacts Fat-Dachsous signaling by increasing the affinity of Fat for Dachsous and decreasing the affinity of Dachsous for Fat[33,34]. The Fj recognition motif (D-X-N-D-[X]$_7$-S/T) is present in multiple EC domains of Drosophila Fat, although the effect on binding has been attributed mostly to phosphorylation of EC3 on Fat[33] and Dachsous[34]. The Fj recognition motif is also conserved in vertebrate Fat and Dchs homologs, but the role of phosphorylation in non-Drosophila organisms is unclear[35]. There is a putative vertebrate homolog of Four-jointed, named Four-Jointed Box 1 (FJX1); however, FJX1 was unable to phosphorylate human Fat1 at Fj recognition motifs in the ECD of Fat1[36].

Here, we determine the structure of a Fat4:Dchs1 complex, revealing several new insights into the unusual heterophilic binding mode of Fat and Dachsous proteins. To complement our structural analysis, we use cellular, computational, and biophysical approaches to study how Fat4:Dchs1 engage in high-affinity interactions and how extracellular phosphorylation modulates Fat–Dachsous binding. Collectively, our data shed new light on the molecular mechanisms by which the atypical Fat and Dachsous cadherins organize tissues and maintain PCP.

## Results

### Mapping Fat4 and Dachsous1 binding domains

In a biophysical binding assay, it was previously shown that the EC1-16 domains of Fat4 and Dchs1 bind with similar affinity to shorter constructs containing only EC1-4[29]. We used isothermal titration calorimetry (ITC) to determine Fat4:Dchs1 binding affinity and further define which domains are essential for their interactions. We purified truncated Fat4 and Dchs1 ECD constructs containing the N-terminal EC1-3 or EC1-4 domains and used ITC to measure their binding affinity. We determined that Fat4(EC1-4) bound to Dchs1(EC1-4) with a dissociation constant ($K_D$) of $0.47 \pm 0.13\,\mu M$, and that Fat4(EC1-4) bound to the shorter Dchs1(EC1-3) construct with a $K_D$ of $0.98 \pm 0.12\,\mu M$ (Fig. 1C) (Supplementary Fig. 6). On the other hand, Fat4(EC1-3) did not detectably bind to Dchs1(EC1-4) (Fig. 1C). We also measured the binding of the truncated Fat4 and Dchs1 constructs to Dchs1 or Fat4-expressing cells, respectively (Supplementary Fig. 1). In agreement with our ITC data, both Dchs1(EC1-3) and Dchs1(EC1-4) constructs bound to the Fat4 cells. In the reverse orientation, Fat4(EC1-4), but not Fat4(EC1-3) bound to the Dchs1 cells. Thus, the EC4 domain of Fat4 is indispensable for binding to Dchs1 while the EC4 of Dchs1 makes a minor contribution to the interaction.

### Structure of the Fat4–Dchs1 complex

We attempted to crystallize several truncated Fat4–Dchs1 complexes to determine the structural basis for Fat4–Dchs1 interactions (Supplementary Fig. 2). Crystals of Fat4(EC1-4) bound to Dchs1(EC1-4) diffracted 3.7 Å resolution, and crystals of Fat4(EC1-4) bound to Dchs1(EC1-3) diffracted 2.3 Å resolution (Supplementary Table 1). The higher-resolution Fat4(EC1-4):Dchs1(EC1-3) structure was solved by molecular replacement, and the resulting model was used to solve the 3.7 Å structure of Fat4(EC1-4):Dchs1(EC1-4). We used the higher-resolution Fat4(EC1-4):Dchs1(EC1-3) structure for all subsequent analyses of interface contacts except for those involving EC4 of Dachsous1.

The Fat4(EC1-4):Dchs1(EC1-4) structure revealed a head-to-tail arrangement with an extended interface that spans multiple domains of each protein (Fig. 2A). The Fat4 EC1–EC4 domains are positioned opposite the Dchs1 EC4, EC3, EC2, and EC1 domains, respectively (Fig. 2A). There were direct contacts between Fat4(EC1):Dchs1(EC4), Fat4(EC2):Dchs1(EC3), and Fat4(EC4):Dchs1(EC1), but no substantial contacts were observed between Fat4(EC3) and Dchs1(EC2). In both Fat4 and Dchs1, the EC domains adopted prototypical cadherin folds and three Ca²⁺ ions could be resolved within interdomain transitions at most of the predicted calcium-binding positions. The Fat4(EC1-4):Dchs1(EC1-4) and the Fat4(EC1-4):Dchs1(EC1-3) structures superimposed with a root mean squared deviation (r.m.s.d) of 1.75 Å (Supplementary Fig. 3), suggesting that they exhibit limited interdomain flexibility.

The Fat4(EC4):Dchs1(EC1) interface contains a hydrophobic cluster centered on Leu379 of Fat4(EC4) and Val120 of Dchs1(EC1). Within this cluster, Fat4 contributes a majority of hydrophobic residues including Val376, Leu417, and Leu419, which contact the methyl groups of Thr118 and Thr126 of Dchs1. On the Dchs1 side, the main contacts are formed by the sidechain of Phe79 and the main chain of β-strand 3. Additional backbone hydrogen bonding interactions can be seen flanking the hydrophobic cluster between the amine of Val376 and the carbonyl of Pro122, between Thr383 and the amine of Glu84, from the epsilon nitrogen of Arg356 to the carbonyl of I80, and from Gln83 to the carbonyl of Val382 (Fig. 2B). The Fat4(EC1):Dchs1(EC4) interface was oriented similarly to the Fat4(EC4):Dchs1(EC1) interface with one notable difference. The equivalent position of Leu379 for Fat4(EC4) is Arg387 in Dchs1(EC4), and this arginine forms a salt bridge with Glu120. (Fig. 2D). The presence of an arginine at this position makes the Fat4(EC1):Dchs1(EC4) interface less hydrophobic than the Fat4(EC4):Dchs1(EC1) interface, and we speculate that this difference

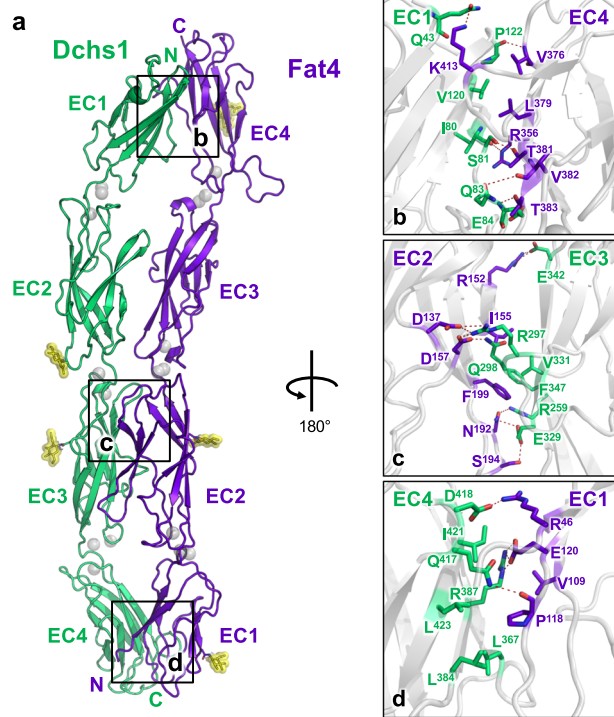

**Fig. 2 | Structure of the Fat4:Dchs1 complex. a** Crystal structure of Fat4(EC1-4) bound to Dchs1(EC1-4) in cartoon representation. N-linked glycans are colored yellow and Ca²⁺ ions are depicted as gray spheres. **b–d**. Zoom panels depicting the Fat4(EC4):Dchs1(EC1) interface (**b**), the Fat4(EC2):Dchs1(EC3) interface (**c**), and the Fat4(EC1):Dchs1(EC4) interface (**d**).

may be linked to the reduced contribution of Fat4(EC1):Dchs1(EC4) to the binding affinity (Fig. 1C).

The EC2:EC3 interfaces are largely hydrophilic and are formed between the β6 and β7 strands of the respective EC3 domains (Fig. 2C). A small hydrophobic core can be found in the Fat4(EC2):Dchs1(EC3) interface containing anchors Fat4-Phe199 and Dchs1-Phe347, in addition to Fat4-Ile155 and Dchs1-Val331. Extensive hydrogen bond networks can be found centered around Arg297 in the β3–β4 loop of Dchs1(EC3) and Asn152 located in the β4–β5 loop of Fat4(EC2). By contrast the Fat4(EC3):Dchs1(EC2) interface is devoid of any close contacts. Although the interface buries a small amount (227 Å²) of surface area, the atomic contacts approach the limit acceptable as weak vdW contacts (4–6 Å).

We introduced several mutations in the Fat4:Dchs1 binding interface to validate our structural model (Supplementary Fig. 4) (Supplementary Table 6). An F358A mutation in Fat4(EC4) and an F79A mutation in Dchs1(EC1) each led to modest (two to fourfold) reductions in binding affinity. A salt-bridge disrupting R152A mutation in Fat4(EC2) led to a stronger decrease (ninefold) in binding affinity, whereas a salt-bridge disrupting R297A mutation in Dchs1(EC2) reduced binding to nearly undetectable levels. An L379R mutation in Fat4(EC4), which replaces a hydrophobic interface leucine with a positively-charged arginine, completely abrogated binding (Fig. 3A, B). Furthermore, we were unable to detect the binding of recombinant Fat4[L379R] tetramers to Dchs1-expressing cells[30] (Supplementary Fig. 5).

We next assessed how the disruptive L379R mutation affects the localization and interactions of Fat4 and Dchs1 in a co-culture cell assay[30]. We used stable HEK293 cells expressing either the Fat4 L379R mutant (Fat4[L379R]) or wildtype Fat4 fused to citrine and co-cultured them with cells expressing Dchs1 fused to mCherry. We then assessed the border accumulation of either Fat4 and Dchs1 proteins, as well as the border complexes formed between Fat4 and Dchs1 (calculated by

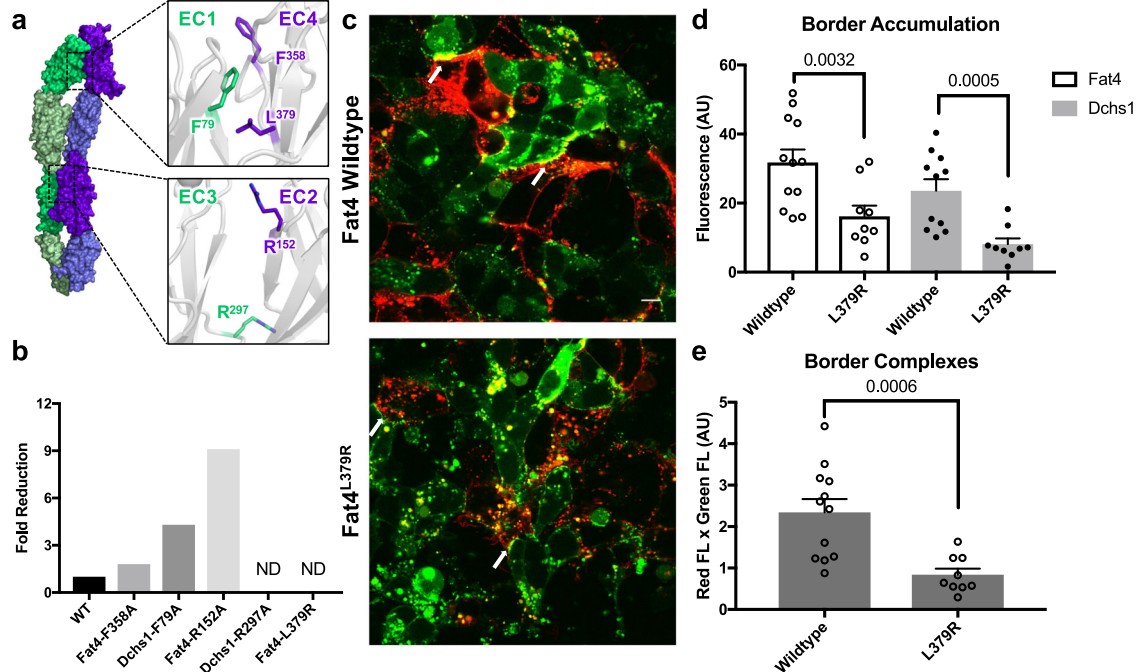

**Fig. 3 | Fat4:Dchs1 interface mutations disrupt binding and cellular adhesion.**
**a** Structural depiction of mutated Fat4 or Dchs1 interface residues. Individual residues are labeled in the zoom panels. **b** Dissociation constants between Fat4 and Dchs1 mutants were determined by ITC and plotted as fold reduction compared to wild-type. Binding was measured between Fat4(EC1–4) and Dchs1(EC1–4) constructs. (ND = not determined) **c** Confocal micrographs (scale bar: 10 μm) depicting co-cultures of Dchs1-expressing cells with either Fat4 or Fat4$^{L379R}$ cells. Border accumulation of Dchs1 and Fat4 proteins are indicated by white arrows. **d, e.** Fluorescence-based quantitation of the border accumulation (**d**) or border complexes (**e**) in co-cultures of Fat4 and Dchs1 or Fat4$^{L379R}$ and Dchs1. $n = 12$ biologically independent cell pairs for wild-type co-cultures and 10 for mutant co-cultures. Data are presented as mean values ± SEM. One-sided unpaired $t$ tests were performed with no correction. $P$-values are noted in the figures. Source data are provided as a Source Data file.

multiplying co-localized fluorescence levels of both proteins). We observed significant reduction in the boundary accumulation of both Fat4, Dchs1, and their complexes in co-culture experiments with Fat4$^{L379R}$ compared to wild-type Fat4 (Fig. 3C–E). This reduction was not due to differences in expression or subcellular localization between Fat4$^{L379R}$ and wildtype Fat4 (Supplementary Fig. 6). Thus, the L379R mutation diminishes both the Dchs1-binding ability and the ability of Fat4 to form complexes.

**Structural comparison of Fat4–Dchs1 interface to cadherin superfamily members**

Structural analysis revealed that the four-domain Fat4:Dchs1 interface closely resembles that of protocadherin homodimers (Fig. 4A). In contrast, the Fat4:Dchs1 structure differs substantially from the single-domain EC1:EC1 interface of classical cadherins homodimers and the two-domain EC1–EC2 "handshake" configuration of CDH23:PDCH15 heterodimers (Fig. 4A). There are four protocadherin subtypes: alpha, beta, gamma, and delta. The structural comparison server DALI identified the γ protocadherin γB3 as the most similar structure to Fat4(EC1–4) and the top hit for Dchs1(EC1–4) was the protocadherin-17 of the δ family (Supplementary Tables 2 and 3). We also compared representative homodimeric structures from the alpha and beta protocadherin families, PDCHα4 and PDCHβ6 (Supplementary Tables 2 and 3) to the Fat4:Dchs1 heterodimer.

The architectures of Fat4:Dchs1 and protocadherin homodimers are similar, with antiparallel binding interfaces spanning along the EC1–EC4 region of each complex. The buried surface area of the Fat4:Dchs1 (2148 Å²) interface was comparable to those of α4 (2159 Å²), β6 (2277 Å²), and substantially larger than γB3 (1535 Å²), and δ17 (1505 Å²) (Fig. 4A). The affinity of Fat4:Dchs1 interactions (0.47 μM) is among the highest reported for known cadherin superfamily members. Additional analysis revealed that Fat4:Dchs1 interact through an

extensive network of 18 hydrogen bonds and 14 salt bridges that may contribute to their high-affinity binding (Supplementary Table 4). This suggests that specific chemical features of the interface, as opposed to the buried surface area, dictate the affinity between both Fat4:Dchs1 and protocadherin homodimers.

The biggest difference between the orientation of the Fat4:Dchs1 domains and those of protocadherins is the positioning of Dchs1(EC1). The alignment of EC2–EC3 of each protein revealed that the EC1 of Dchs1 is tilted by ~35° towards Fat4, making it the only protomer that adopts such a pronounced curvature in a protocadherin-like binding interface (Fig. 4B). This distinctive conformation was observed in both the high- and low-resolution structures (Supplementary Fig. 8A). On the Fat4 side, EC1 is also tilted by a lesser degree (approximately 22°) towards Dchs1 (Fig. 4B). These bent EC1 conformations create a "puckering" effect that helps separate Fat4(EC3)–Dchs1(EC2) and prevents them from forming additional contacts (Fig. 2A, Fig. 4B). To determine if the tilt in Dchs1 is due to structural differences we aligned EC1 for each protocadherin representative (Supplementary Fig. 8B). In the β5–β6 loop there is a disulfide-linked region containing 7 residues, including the cysteines, that is highly conserved across every protocadherin subfamily and makes interdomain contacts with the β2–β3 loop of EC2. This may act as a "staple" to stabilize the calcium-binding linker. These residues are not present in Fat4 and Dchs1, which results in a truncation of four residues for Fat4 and six residues for Dchs1.

We next aligned individual Fat4, Dchs1, and protocadherin EC domains to compare their binding modes and secondary structural features (Fig. 4C) (Supplementary Fig. 7). Fat4 EC1 and EC2 adopt prototypical EC domain folds that are similar to the analogous protocadherin domains except for interface contacts. In Fat4(EC3), the β3–β4 loop was truncated by approximately five residues compared to those of the protocadherin EC3 domains. In protocadherins, the β3–β4 loop contributes to family-specific interactions[37,38]. In Fat4(EC3),

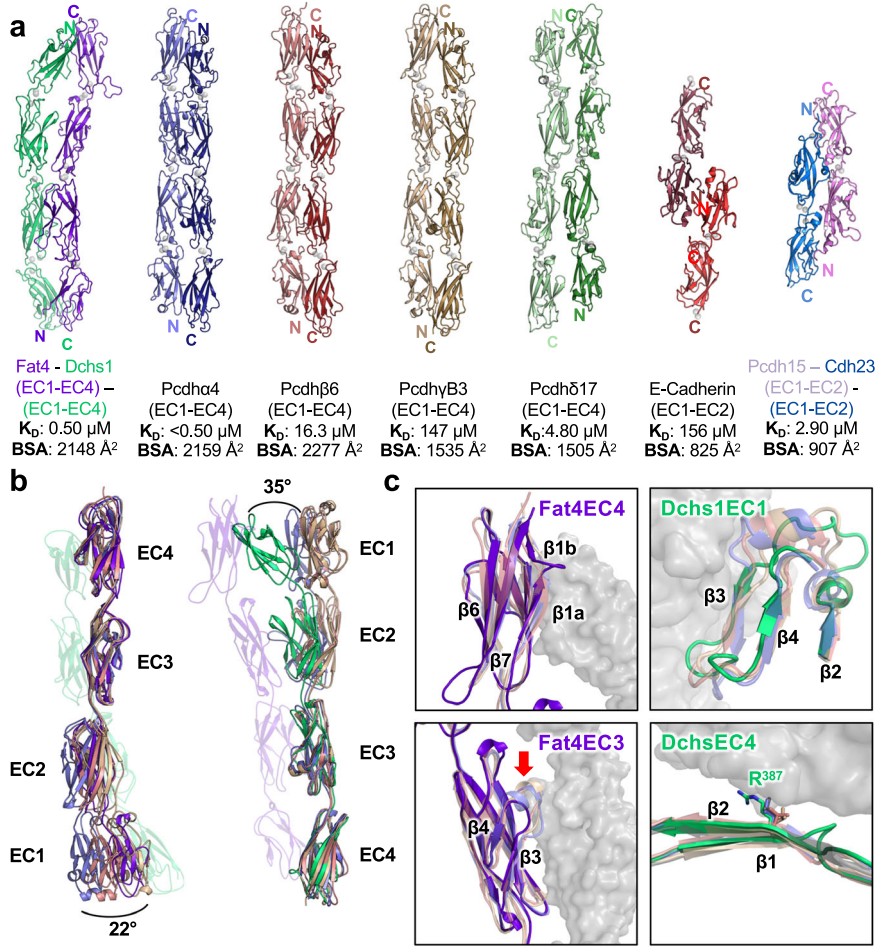

**Fig. 4 | Comparison of Fat4:Dchs1, cadherin, and protocadherin structures.**
**a** Comparison of the Fat4:Dchs1 structure with structures of selected proto-cadherin homodimers, the E-cadherin homodimer, and the PCDH15:CDH23 het-erodimer. The buried surface area and dissociation or homodimerization constants for each interaction are indicated beneath the structures. **b** The EC3-EC4 domains of Fat4 or Dchs1 and several protocadherins were aligned to highlight the tilted orientation of Fat4(EC1) and Dchs1(EC1). **c** Zoom panels showing distinctive structural features in Fat4 or Dchs1 domains. Upper-left: an insertion within the β1 loop of Fat4(EC4) contacts Dch1(EC1). Upper-right: Dchs1(EC1) has less secondary structure content than protocadherin EC1 domains. Bottom-left: the connecting loop between β3 and β4 of Fat4(EC3) is truncated and does not interact with Dchs1(EC2). Bottom-right: a R387 residue in Dchs1(EC4) is not conserved in pro-tocadherins and makes substantial contacts with Fat4(EC1).

truncation of these residues prevents interdomain contacts with Dchs1(EC2), providing a structural mechanism for preferential inter-actions between Dchs1(EC3) and Fat4(EC2). In Fat4(EC4), notable dif-ferences were a "split" β1 strand containing an insertion that contacts Dchs1(EC1) (Fig. 4C) and an extended β6–β7 loop of unknown sig-nificance. This β1 strand insertion is rarely observed and is only present in select members of the δ-protocadherin family. In Dchs1, EC2–4 were structurally similar to EC2–4 of other protocadherin members. How-ever, EC1 lacked secondary structure in the region spanning strands β2–β4. By contrast, alpha, beta, and gamma members form helices between β2 and β3 that orient β3 towards the core of the domain (Fig. 4C). The absence of this helix in Dchs1(EC1) allows β3 to extend outward toward Fat4(EC4) to create additional interface contacts.

### Conservation of the Fat4–Dchs1 binding interface
Fat and Dachsous homologs are encoded by organisms ranging from cnidarians to humans[7,39]. We performed a conservation analysis to investigate whether the binding mode observed in the human Fat4:Dchs1 structure is broadly representative of Fat-Dachsous inter-actions across a panel of species (Supplementary Table 5). Binding interface residues were then painted according to their level of con-servation (Fig. 5A). The residues forming the Fat4(EC1):Dchs1(EC4), Fat4(EC2):Dchs1(EC3), and Fat4(EC4):Dchs1(EC1) interfaces are mostly

conserved across the species sampled. The L379 residue in Fat4(EC4), which is critical for Dchs1 interactions (Fig. 3B) was also conserved and only varies between leucine and valine. Within Dchs1(EC4), the Arg387 residue central to the Fat4-binding interface is highly conserved across all species sampled. On the other hand, opposing residues in the minimally-contacting Fat4(EC3) and Dchs1(EC2) domains are poorly conserved, which is consistent with a lesser role in Fat4:Dchs1 binding (Fig. 2A, Fig. 5A). Taken together, these findings suggest that Fat and Dachsous interact via a similar binding mode in various metazoan organisms.

### Human Fat4 is not efficiently phosphorylated by Four-jointed homologs
Alignment of the EC3 domains from our sampled list of organisms revealed that the Fj phosphorylation motif (D-X-N-D-[X]$_7$-S/T) was fully conserved in Dachsous homologs and mostly conserved in Fat homologs, including human Fat4 (Fig. 5B). Given that phosphorylation of Fat and Dachsous regulates PCP in *Drosophila*[35], we tested whether Fat4 or Dchs1 ECDs are amenable to phosphorylation. We performed an in vitro phosphorylation assay[35] to test whether Fat4(EC1–4) and Dchs1(EC1–4) are modified by *Drosophila* Fj or the putative human ortholog Four-jointed box 1 (FJX1). *Drosophila* Ft(EC1–4) and Ds(EC1–4) were used as positive controls. Following enzymatic

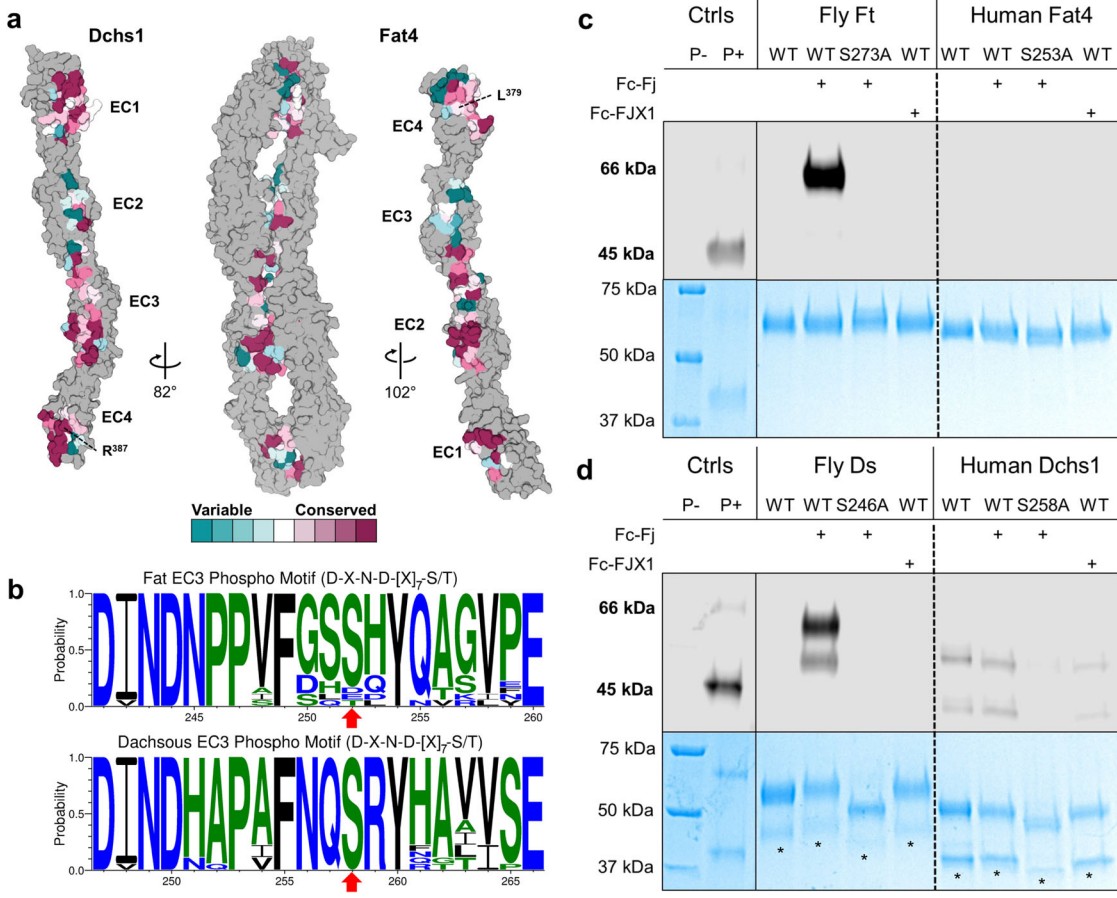

**Fig. 5 | Interface and phosphomotif conservation in Fat and Dachsous.**
**a** Conservation analysis of the Fat4 and Dchs1 binding interface. The EC1-4 domains of each protein are depicted in surface representation and interacting residues are colored based on their conservation across 18 different Fat4 or Dchs1 homologs. **b** A multiple sequence alignment of the Fj phosphorylation motif (D-X-N-D-[X]$_7$-S/T) in EC3 of Fat4 and Dchs1 is shown as a sequence logo. The predicted phosphoserine residue is indicated with a red arrow. The motif is mostly conserved in Fat4 and Dchs1 homologs. **c,d**. SDS-PAGE analysis of Fat4(EC1–4), Dchs1(EC1–4), Ft(EC1–4), and Ds(EC1–4) proteins following incubation with Fj and FJX1. The gels were stained with Coomassie to monitor total protein (bottom panel, colored blue)

and with a phosphostain (top panels, grayscale) to monitor phosphorylated protein levels. The Drosophila Ft and Ds proteins were robustly phosphorylated with Fj but not FJX1, and there was no detectable signal in the phosphomutants. Phosphorylation of human Fat4 was not detected following treatment with Fj or FJX1. Dchs1 was endogenously phosphorylated at low levels, and this phosphorylation signal was diminished in the phosphomutant. Unphosphorylated and phosphorylated protein ladders were used as controls. Asterisks indicate fragmented Dchs1 protein bands that reproducibly arise following treatment with Fj. SDS-PAGE analysis was performed three times using different batches of recombinant protein each time. Source data are provided as a Source Data file.

treatment, the proteins were analyzed by SDS-PAGE, and phosphorylation was assessed using a phosphoprotein stain. We found that *Drosophila* Ft and Ds are modified at the predicted serine positions by Fj but not by FJX1 and that Fat4 was not modified by either enzyme (Fig. 5C, D). Unexpectedly, Dchs1 had a low but detectable phosphorylation signal when purified from *Trichoplusia ni* insect cells regardless of treatment with Fj (Fig. 5D). Mutation of the predicted phosphorylation site in Dchs1 eliminated the phosphorylation signal, suggesting that the *T. ni* cells express a kinase that modifies Dchs1 at the Fj motif (Fig. 5D). We also tested whether Fat4(EC1–4) and Dchs1(EC1–4) were endogenously phosphorylated when expressed in mammalian cells, potentially by an unidentified kinase. However, we found that neither Fat4 nor Dchs1 was modified when expressed in HEK293 cells (Supplementary Fig. 10). The inability of Fj to phosphorylate Fat4 indicates that there are additional sequence requirements beyond the known D-X-N-D-[X]$_7$-S/T motif, and the general lack of phosphorylation in Fat4 and Dchs1 suggests that this regulatory mechanism was not conserved in mammals.

## Phosphoregulation of Fat-Ds interactions in *Drosophila*

Phosphorylation reportedly influences *Drosophila* Fat-Dachsous function by modulating their binding affinity[33,34]. Using the

Fat4:Dchs1 structure as a guide, we modeled the *Drosophila* Ft(EC1–4):Ds(EC1–4) complex to predict how phosphoserines affect the binding interface. We generated Alphafold models[40] of the complex and found that the fourth highest scoring model was similar to the experimentally determined Fat4:Dchs1 structure, whereas the top three hits did not predict meaningful interactions. The models do not contain $Ca^{2+}$ ions and therefore may not represent the energy minimized configuration. Nevertheless, the resultant *Drosophila* model closely matched the orientation of the calcium-bound human Fat4:Dchs1 complex and the phosphate-bearing EC2–EC3 regions superimposed with an RMSD of 1.79 Å (Supplementary Fig. 9).

When phosphoserines were introduced at the predicted EC3 positions in Ft and Ds, each was situated near the binding interface. The Ft phosphate was positioned within 5 Å of the positively-charged guanidinium group of Ds-Arg177, and the Ds phosphate was approximately 6.3 Å from the hydroxyl group of Ft-Thr210 (Fig. 6A). Further, the majority of residues modeled in the Ft(EC2) β4–β5 loop that contains Thr210 are negatively charged. This positioning suggests that phosphorylation of Ft can strengthen Ft:Ds interactions by introducing polar contacts across the Ft(EC3):Ds(EC2) binding interface, likely by generating a salt bridge with Arg177 of Ds. Meanwhile, phosphorylation of Ds can introduce repulsive forces into the Ft(EC2):Ds(EC3)

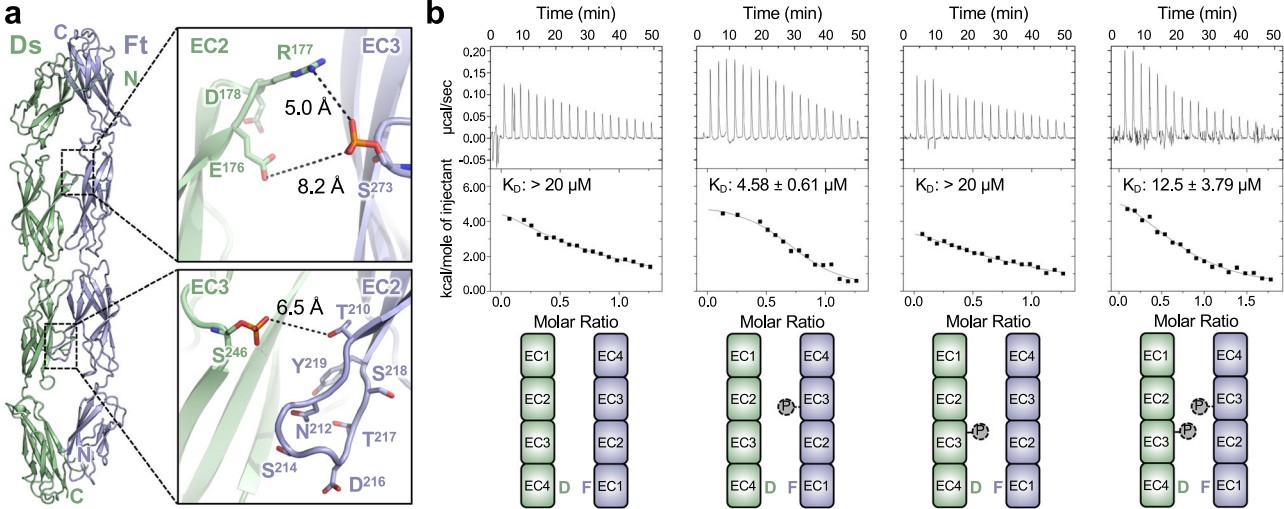

**Fig. 6 | Phosphorylation of *Drosophila* Ft enhances binding to Ds. a** Alphafold model of *Drosophila* Ft(EC1–4) bound to Ds(EC1–4) without Calcium ions. Modeled phosphoserine modifications (colored orange) in EC3 oppose polar or charged residues at the interface. **b** Representative ITC thermograms depicting interactions between *Drosophila* Ft(EC1–4) and Ds(EC1–4), phosphorylated Ft(EC1–4) and unmodified Ds(EC1–4), unmodified Ft(EC1–4) and phosphorylated Ds(EC1–4), and phosphorylated Ft(EC1–4) and phosphorylated Ds(EC1–4). The thermograms were fitted to determine a $K_D$ of 4.58 μM between phosphorylated Ft and non-phosphorylated Ds, and a $K_D$ of 12.5 μM between phosphorylated Ft and phosphorylated Ds. The affinities of non-phosphorylated Ft for either non-phosphorylated Ds or phosphorylated Ds were too low to be detected. $K_D$ measurements are an average of $n = 2$ independent experiments. Source data are provided as a Source Data file.

interface which may destabilize the interaction. Although the predicted interactions are not close contacts (≥5 Å), we anticipate that the lack of calcium ions in the model, or fluctuations in domain orientations could account for this disparity. For example, EC2–EC3 contacts in protocadherins have been found to "break" in molecular dynamics simulations, and a similar disengagement in Fat4–Dchs1 could push the Fj-modified phosphoserines into contact with interface residues[48].

To complement our modeling, we used ITC to directly assess the effect of phosphorylation on Ft:Ds affinity. We compared the binding between unmodified *Drosophila* Ft(EC1–4) and Ds(EC1–4), phosphorylated Ft(EC1–4) and unmodified Ds(EC1–4), unmodified Ft(EC1–4) and phosphorylated Ds(EC1–4), and phosphorylated Ft(EC1–4) and phosphorylated Ds(EC1–4) (Fig. 6B) (Supplementary Table 6). We found that the affinities between unmodified Ft and unmodified or phosphorylated Ds were too low to be effectively determined by ITC. However, phosphorylation of Ft enhanced binding to unmodified Dachsous and the $K_D$ was determined to be $4.58 \pm 0.61$ μM. This affinity was reduced to $12.5 \pm 3.79$ μM when Ds was phosphorylated, confirming the putative destabilizing effect of Ds modification. This indicates that Fj-mediated phosphorylation plays a direct role in regulating Fat–Dachsous interactions by tuning binding affinity within *Drosophila* Ft:Ds.

## Discussion

Our structural and biophysical studies provided several insights into the molecular function of Fat and Dachsous proteins. We determined that EC1–4 of Fat4 and Dchs1 interact in a manner that resembles protocadherin homodimers. Along with their inherent flexibility, this binding mode enables Fat4 and Dchs1 to remain bound over a wide range of intercellular spaces (~40–120 nm)[29,30] and may be important for maintaining cell polarity during periods of rapid growth during development. Intercellular tethering has been shown to preserve topological information within forming tissues in planarians[41] and may ensure that cells remain clustered during morphogenetic events such as collective migration[42] or valve formation[43,44]. ITC measurements indicated that Fat4–Dchs1 bind with higher affinity than classical cadherins and most protocadherins, which may enable them to maintain contacts at cell boundaries under a variety of strenuous conditions

(e.g., shear stress or cell motility). Clustering[30,45] and dimerization[29] of Fat and Dachsous proteins may also strengthen the avidity of their interactions, suggesting that additional structural features may influence their ability to organize tissues.

The kinked orientation of Dchs1 EC1 relative to EC2 and subtle differences in Fat4 EC3 and EC4 domain structure contribute to the asymmetry of Fat4–Dchs1 compared to the symmetrical protocadherin dimers. Additionally, an extensive network of fourteen salt bridges enables Fat4:Dchs1 to interact with high affinity ($K_D = 0.47$ μM). The four-domain binding mode of Fat4:Dchs1 contrasts with the EC1–EC2 "handshake" conformation of PDCH15:CDH23, which, to our knowledge, represents the only other structure of a heterophilic cadherin complex[31]. Despite this smaller binding interface and moderate affinity ($K_D = 2.9$ μM), the structure of PDCH15:CDH23 is optimized such that they remain bound and convey forces from sound waves to hair-cell transduction channels to affect hearing[46]. Given that Fat and Dachsous PCP may be regulated by tension anisotropy[47], it will be interesting to investigate whether mechanical forces can be similarly withstood by the Fat–Dachsous interaction.

Extracellular phosphorylation is a distinctive feature of Fat proteins and regulates PCP in *Drosophila*. Together with our biochemical data, structural modeling suggests that phosphorylation of Fat enhances Fat-Dachsous affinity by generating polar contacts in a region that is otherwise devoid of interactions. On the other hand, phosphorylation of Dachsous appears to have a destabilizing effect in a region containing multiple well-conserved interface residues. Given that protocadherins cluster and are conformationally dynamic within each individual interface[48], the ability of Fat/Ds to modify individual interfaces could conceivably fine-tune overall binding affinity to accommodate the requirements of cells within a specific tissue. This type of context-dependent binding affinity modulation resembles the glycan-mediated enhancement of Notch receptor–ligand binding[49,50] and highlights a broad role in the posttranslational regulation of extracellular interactions in development.

Recombinant human Fat4 was not phosphorylated by Fj or the human homolog FJX1, and Fat4(EC1–4) was not phosphorylated regardless of whether it was expressed in human or insect cells. Given that Fat4(EC3) contains the Fj phosphomotif, we note that Fj-mediated

phosphorylation of EC domains must have additional structural requirements beyond the established D-X-N-D-[X]$_7$-S/T sequence. Interestingly, if the phosphomotif were assumed to be the sole requirement for Fj-mediated phosphorylation, it is possible that other cadherins could be modified by Fj. For example, the adhesion GPCR Flamingo (CELSR in vertebrates) is a member of the core PCP pathway and contains the Fj phosphomotif in EC repeats 4 and 6, both of which are conserved between flies and vertebrates. It was previously reported that several EC domains of Fat1 were phosphorylated, but that FJX1 was incapable of phosphorylating the Fat1 protein[36]. These data coincide with our findings that FJX1 does not phosphorylate Fat4. It is currently unknown if the Fat4 ectodomain is phosphorylated by any mammalian kinase, but our data confirm that Fat4 is not phosphorylated by FJX1. Thus, it remains unclear whether the functional or regulatory role of Fat–Dachsous phosphorylation is conserved outside of the *Drosophila* system.

Fat proteins are multifunctional and appear to have distinct Dachsous-dependent and ligand-independent functions. For example, Fat-Dachsous interactions regulate PCP, the Fat4 ICD modulates Hippo signaling by engaging the scaffold protein Angiomotin-like 1 (Amotl1), and Fat4 inhibits RET[51] signaling to guide kidney development. Additionally, nearly all cancer-associated Fat4 and Dchs1 mutations in the COSMIC database are located outside of the binding interface, suggesting that the tumor suppressor role of Fat and Dachsous may not require their direct interactions. Through our mutational studies, we isolated and characterized null-binding L379R and affinity-reducing R297A mutants in Fat4 and Dchs1, respectively. In the future, we anticipate that in vitro or in vivo assessment of these mutations will reveal biological functions that specifically rely on Fat4–Dchs1 binding.

## Methods

### Protein expression and purification

Human Fat4 truncations (EC1–EC3 aa43–353, EC1–EC4 aa43–475) were cloned into the pAcgp67a vector with a N-terminal gp67 leader sequence followed by residues Asp-Pro, and containing a C-terminal 3C protease site followed by biotin acceptor peptide (BAP-tag: GLNDIFEAQKIEW) and 6×-His tag. Human Dchs1 truncations (EC1–EC3 aa43–362, EC1–EC4 aa43–472) were cloned into the pAcgp67a vector with a N-terminal gp67 leader sequence followed by residues Asp-Pro, and containing a C-terminal 3C protease site followed by biotin acceptor peptide (BAP-tag: GLNDIFEAQKIEW) and 6×-His tag.

All proteins were expressed using baculovirus by infecting Trichoplusia Ni cells in ESF921 Insect Cell Culture Media (Expression Systems) at a density of $2 \times 10^6$ cells/mL and harvesting culture after 72 h. Proteins were purified using nickel affinity and size exclusion chromatography.

### Site-directed mutagenesis

Site-directed mutagenesis was performed by amplifying Fat4(EC1–4) and Dchs1(EC1–3), using internal primers to perform the relevant nucleotide mutation and Gibson Assembly to ligate the two inserts and the pAcgp67a vector together. Internal primers are described in Supplementary Table 7, with the mutated nucleotides in bold. Proteins were produced as described above.

### Proteolytic processing

Fat4 and Dchs1 constructs intended for co-crystallization were purified individually as described above and enzymatically processed to cleave C-terminal tags. All tags were cleaved using 1:250 (w/w) 3C protease, 1:100 (w/w) carboxypeptidase A (Sigma), and 1:100 (w/w) carboxypeptidase B (EMD Millipore). Proteins were then purified individually by size-exclusion chromatography in 20 mM HEPES buffer pH 7.4 containing 50 mM NaCl and 1 mM Calcium Chloride (low-salt HBS + Calcium). Fractions corresponding to monomers were pooled. Fat4 and Dchs1 truncations were mixed in a 1:1 stoichiometry and purified

again by size-exclusion chromatography, and fractions corresponding to the co-eluted complex were pooled.

### Crystallization

Proteolytically processed Fat4(EC1–4)-Dchs1(EC1–3) complexes were concentrated to ~25 mg/mL as measured by Nanodrop in low-salt HBS + Calcium and crystallized in a sitting drop using the vapor diffusion method. Drops containing 300 nL of protein were combined with 100 nL of seed stock and 200 nL of mother liquor consisting of 0.08 M MgCl$_2$, 0.1 M Sodium Cacodylate, pH 6.1, and 20% polyethylene glycol (PEG) 1000. Crystals were harvested and cryoprotected using mother liquor + 30% Ethylene Glycol in a stepwise manner.

Fat4(EC1-4)-Dchs1(EC1-4) complexes were proteolytically processed and concentrated to 10 mg/mL as measured by Nanodrop in low-salt HBS + Calcium and crystallized in a sitting drop using the vapor diffusion method. Final hits were obtained by streak seeding into 0.1 M MES, pH 6.5, 0.2 M NaCl, and 10% PEG 4000. Crystals were harvested and cryoprotected using mother liquor + 30% ethylene glycol directly.

### Data collection and structure determination

Data for both Fat4(EC1–4)-Dchs1(EC1–4) and Fat4(EC1–4)-Dchs1(EC1–3) crystals were collected at Advanced Photon Source beamline 22-BM. Data were indexed, integrated, scaled and merged with XDS. Fat4(EC1–4) and Dchs1(1–3) crystallized in space group P12$_1$1 with unit cell dimensions $a = 88.7$ Å, $b = 61.4$ Å, $c = 101.0$ Å, $\beta = 112.7$ Å and one complex per asymmetric unit. The structure of the Fat4(EC1–4)−Dchs1(EC1–3) complex was solved using molecular replacement (MR). Models for each domain were obtained from various previously solved structures (PDBIDs: 2A4C, 4ZPL, 4ZI8, 4ZPO, 5IU9, 3Q2W). MR was performed in PHENIX[52] using Phaser. Iterative rounds of MR were performed using all seven homology models, inspecting the result for fitting after each round. Once six out of seven domains were placed, the respective domain sequence was threaded onto the homology model and MR repeated with the new threaded models. For Fat4 EC4, an initial homology model was used to locate secondary structures and then a polyalanine model was built into the density, followed by manual building to correct the domain sequence. Fat4 EC1 and Dchs1 EC1 also had to be rebuilt manually. The model was initially subjected to rigid body refinement, and then was subjected to several rounds of positional, TLS, and B-factor refinement using PHENIX_refine. Manual building was performed in COOT. The final structure has a $R_{work}$ of 19.01% and a $R_{free}$ of 23.57%. The structure contains Fat4 residues 43–468 and Dchs1 residues 43–354.

Fat4(EC1–4) and Dchs1(EC1–4) crystallized in space group P222 with unit cell dimensions $a = 64.222$ Å, $b = 82.102$ Å, $c = 224.759$ Å and one complex per asymmetric unit. The structure was solved using the Fat4(EC1–4)−Dchs1(EC1–3) structure and domain 4 of PDB model 6VFT using molecular replacement, followed by manual rebuilding of Dchs1 EC4 and several rounds of positional and B-factor refinement using ISOLDE and PHENIX_refine. Manual building was performed in COOT. The final structure has a $R_{work}$ of 27.09% and a $R_{free}$ of 29.78%. The structure contains Fat4 residues 43–467 and Dchs1 residues 43–462. There is also an additional Proline at the N-terminus from the expression construct.

### ITC binding studies

ITC experiments were performed using a MicroCal ITC$_{200}$ calorimeter and designed following the manufacturer's manual. Wild type, F358A, R152A, and L379R Fat4(EC1–4) and wild type Fat4(EC1–3) were produced as described above, concentrated to 30 μM, and placed in the sample cell. Wild type, F79A, and R297A Dchs1(EC1–4) and wild type Dchs1(EC1–3) were produced as described above, concentrated to 300 μM, and used as titrants. Concentrations were determined using sample absorbance at

280 nm and theoretical extinction coefficients using the Nanodrop spectrophotometer. All experiments were performed at 20 °C. A typical experiment consisted of an initial 0.4 μL injection, followed by nineteen 2.0 μL injections of titrant ($\Delta t = 150$ s). All experiments were repeated at least once. Fittings and binding constants were obtained using Origin software and modeled using one set of sites.

ITC experiments to assess the effect of phosphorylation on fly Ft(EC1–4) and Ds(EC1–4) followed the same protocol as outlined above. Proteins were produced using insect cells, modified as needed according to the protocol illustrated under Phosphorylation studies (see below), polished using size exclusion chromatography, and concentrated to ~50 μM and ~300 μM, respectively, as determined by sample absorbance at 280 nm using the Nanodrop spectrophotometer and theoretical extinction coefficients determined using ProtParam. Experiments were performed using the same conditions as above. Fittings and binding constant ranges were obtained using Origin software and modeled using one set of sites.

## Structural analysis of Fat4·Dchs1 complex

Model analysis was done using Pymol[53–55]. Structural searches to identify proteins with similar architectures were done using the DALI server[56]. Shape complementarity determinations were done using ccp4[57–59]. Buried surface area analysis was done using the PISA server[60]. Multiple species sequence alignments were pulled from Uniprot[61] and organized using Jalview[62]. Conservation painting was done using the ConSurf server[63–67]. Phosphoserine modeling was done using Coot[68].

## Phosphorylation studies

N-terminally Fc-tagged fly Four-jointed (Fj) containing residues 100–583 and N-terminally Fc-tagged human Four-jointed box 1 protein (FJX1) containing residues 118–437 were produced in insect cells as described for production of recombinant Fat and Dachsous proteins.

Recombinant fly Ft(EC1–4), fly Ds(EC1–4), human Fat4(EC1–4), and human Dchs1(EC1–4) were phosphorylated in vitro in 10 μL reactions containing 0.1 mM ATP, 50 mM Tris, pH 7.0, 10 mM Manganese Chloride, 100 mM Calcium Chloride, 0.17 μM Fj or FJX1, and 3.58 μM substrate. Reactions were incubated at 27 °C for Fj-mediated phosphorylation and 37 °C for FJX1-mediated phosphorylation for 1 h. Samples were then diluted 1:1 with reduced loading buffer, boiled, and ran on a 15% acrylamide gel (Bio-Rad). Phosphorylation was detected using the ProQ Diamond Phosphoprotein stain (Invitrogen) as per manufacturer's instructions and visualized on a Licor Odyssey Fc imager using a 2 min integration time on the 600 channel. Gels were then stained using Coomassie Blue and imaged using a Bio-Rad GelDoc XR+.

Full-length FJX1 was cloned into pVLAD6. Human Fat4(EC1–4) and fly Ft(EC1–4) were cloned into pVLAD6L containing an IgG signal peptide. Human Dchs1(EC1–4) and fly Ds(EC1–4) were cloned into pVLAD6 with a gp67a signal peptide. All constructs were used to produce BacMAM viruses which were then used to infect HEK293-ES suspension cells (Expression Systems).

Briefly, 4 mL of cells at a concentration of $1.5e^6$ cells/mL were seeded into 12-well plates and infected with BacMAM virus for each Fat or Dachsous protein alone and in combination with FJX1 BacMAM virus. Wells were supplemented with Sodium Butyrate at final concentrations of 10 mM. After 2–3 days, the wells were collected and the supernatant separated from the cell pellet. The supernatant was incubated with Ni-NTA resin for 4 h, washed with HB(50 mM) S + 1 mM CaCl$_2$ + 10 mM Imidazole, and eluted using 20 μL HB(50 mM)S + 1 mM CaCl$_2$ + 500 mM Imidazole. Eluate was diluted 1:1 using reducing loading buffer, boiled, and ran on a 15% acrylamide gel (Bio-Rad). Phosphorylation and protein levels were detected as mentioned above. Cell pellets were diluted into 100 μL reduced loading buffer and boiled. Dilutions were ran on a 15% acrylamide gel, transferred onto PVDF using the iBlot 2 (Thermo-Fisher)

transfer apparatus. Membrane was blocked, incubated with anti-His-tag antibody (1:10,000 v/v) (Bethyl Laboratories), and visualized using the Pierce ECL Western Blotting substrate (Thermo-Fisher) and a Licor Odyssey Fc imager using the Chemiluminescence setting.

## Confocal microscopy

Cell lines were grown in adherent cultures in Dulbecco's Modified Eagle's Medium (DMEM) supplemented with 10% FBS. Cells were cultured in a humidified atmosphere of 5% CO$_2$ at 37 °C.

HEK293 cells transduced with Fat4-mCitrine and Dachsous1-mCherry (courtesy of the David Sprinzak lab) were seeded onto 24-well glass bottom plates (Cellvis) treated with 100 μg/mL Concanavalin A (Millipore Sigma) and allowed to adhere for 3 days. For Fat4-mCitrine cells were washed using low-salt HBS + 1 mM CaCl$_2$ + 1% BSA. Tetramers were prepared by incubating C-terminally biotinylated Dchs1(EC1–4) with Streptavidin-AF647 (produced in-house) for 10 min on ice. Tetramers were added to wells and incubated at 37 °C for 1 h. Cells were then washed once, fixed using 2% formaldehyde for 15 min at room temperature, and the solution diluted 1:3 before image collection. Dachsous1-mCherry cells were allowed to adhere for 3 days, then induced using Doxycycline at a concentration of 100 ng/mL for 1–2 days before staining with tetramers of Fat4 incubated with Streptavidin-AF488 (Invitrogen) (prepared as described above). Images were taken using a Keyence BZ-X710 confocal microscope. Analysis was performed using ImageJ.

A co-culture of HEK293-Dchs1 mCherry WT cells with HEK293-Fat4-mCitrine WT cells or HEK293-Fat4-mCitrine mutant (L379R) cells were seeded onto 24-well glass bottom plates (Cellvis, USA). Directly prior to imaging the media was replaced with low fluorescence imaging media (αMEM without Phenol red, ribonucleosides, deoxyribonucleosides, folic acid, biotin and vitamin B12 (Biological Industries, Israel).

Coculture experiments were imaged using a Zeiss LSM 880 confocal microscope using a 63× objective. To estimate the colocalization of FAT4 and Dchs1 on the boundaries between the cells, we first manually traced the boundaries using a custom made Matlab code. To create a binary mask that represents the boundaries region, we dilated the manually delineated lines by a radius that is slightly larger than the characteristic boundary width (this was estimated by analyzing several representative images). This edge mask was used to separately estimate the amount of FAT4 and Dchs1 fluorescence on the boundaries. To estimate the colocalization on the boundaries we applied Gaussian blurring on both channels. The blurred channels were multiplied by each other, then by the edge mask, and then summed and divided by the total length of the boundaries. Graphpad Prism was used for data visualization and analysis.

## Flow cytometry

HEK293 cells were seeded in 24-well plates at approximately 70% confluence 24 h before FACS. Directly prior to FACS, cells were trypsinized, spun at 1000 rpm for 5 min, and resuspended in 200 μL of FACS buffer. FACS buffer consisted of PBS with 1% FBS and 5 mM EDTA. Flow cytometry was performed using a Cytoflex5L flow cytometer (Beckman Coulter). The gating strategy used can be found in Supplementary Fig. 12. Kaluza software was used to analyze the data.

## Structural modeling

Alphafold models were developed by fusing *Drosophila* Fat (residues 66–494) to *Drosophila* Dachsous (residues 22–450) using a 15× Glycine linker and submitting to ColabFold[69]. Out of the five models produced, models 4 and 5 were the only results that contained a binding interface resembling the Fat4–Dchs1 complex. Model 4 was used for analysis.

## Reporting summary

Further information on research design is available in the Nature Portfolio Reporting Summary linked to this article.

## Data availability

The crystallography data generated in this study have been deposited in the Protein Data Bank under the accession code 8EGW (Fat4(EC1–4):Dchs1(EC1–3)) and 8EGX (Fat4(EC1–4):Dchs1(EC1–4)). Source data are provided with this paper.

## Code availability

The MATLAB code used to analyze the co-culture experiments can be found at https://github.com/dsprinzak/Medina-et-al.

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

## Acknowledgements

We thank the staff at the 22-BM beamline of the Advanced Photon Source for assistance with remote X-ray data collection We also thank T. Tran from Moffitt Chemical Biology Core for helping with ITC studies and Q. Ming and D. Antfolk for helpful suggestions. This project was supported by NIH R35GM133482 (V.C.L.), and NIH R35 Diversity Supplement R35GM133482-03S2 (E.M.), and the Israeli Science Foundation (grant No. 1388/18) (Y.E. and D.S.). V.C.L. is a Rita Allen Scholar. Support for shared resources was provided by the Moffitt Cancer Center Support grant NIH P30CA076292.

## Author contributions

V.C.L., E.M., and D.S. designed the experiments. E.M. performed the protein purifications, crystallography, binding studies, and computational modeling. E.M., Y.E., D.S., and D.L. performed the microscopy imaging. Y.E. performed co-culture assays. V.C.L. and E.M. wrote the paper. D.S., E.K.L., E.M., and V.C.L. edited and reviewed the paper. V.C.L. supervised the research.

## Competing interests

V.C.L. is a consultant on an unrelated project for Cellestia Biotech. The remaining authors declare no competing interests.
