## [Peer Review File · Nature Communications]

Structure of the planar cell polarity cadherins Fat4 and Dachshous1REVIEWER COMMENTS

Reviewer #1 (Remarks to the Author):

The authors present novel and valuable data on the structure and strength of the heterophilic binding between human Fat4 and Dchs1, and extend this to show likely parallels with their *Drosophila* homologs Fat and Dachso1. The authors largely limit their analysis to the N-terminal 4 cadherin (EC1-4) repeats of these proteins, as these had been previously shown to be sufficient for binding, although to my knowledge no one has tested their necessity. The authors also take up the issue of phosphorylation sites in EC3 of these proteins, where it lies in the binding domains and how phosphorylation improves the binding of *Drosophila* Fat to Dachso1.

The structural and binding data all look quite valuable, but outside my expertise, so I will limit most of my questions to the phosphorylation studies. Work on *Drosophila* Fat and Dachso1 identified sites in specific ECs as targets of the Golgi-resident kinase Fj, and argued through mutation experiments that phosphorylation at EC3 improved Fat's affinity for Dachso1, while phosphorylation of EC3,6 and/or 9 reduced Dachso1's affinity for Fat (Ishikawa et al., 2008; Brittle et al., 2010; Simon et al., 2010).

The authors confirm part of this by showing that Fat(EC1-4), when phosphorylated in vitro by Fj, binds more strongly to Dachso1(EC1-4). However, the authors did not completely test the effects of Dachso1 phosphorylation. While phosphorylated and non-phosphorylated Dachso1(EC1-4) bound unphosphorylated Fat(EC1-4) too weakly to detect a difference, I wonder why the authors did not test whether phosphorylation of Dachso1(EC1-4) changed its affinity for phosphorylated Fat(EC1-4)? Nor did the authors discuss possible effects on the binding structure that would be predicted from phosphorylation of Dachso1 EC3, and why this might lead to weaker binding.

There is also one prior result that might be worth discussing, as it suggests that phosphorylation of domains other than the EC3 might modify binding: Simon et al. claimed that Fj coexpression still increased the affinity of Fat for Dachso1 even when the Fat lacked its EC3 phosphorylation site, albeit not as strongly as with wild type Fat (see their Fig. S3). Could the authors comment briefly on the possible roles of the other EC Fj targets?

More difficult is drawing parallels with Fat4 and Dchs1, as the authors have not detected any phosphorylation of Fat4(EC1-4), and weak phosphorylation of Dchs1(EC1-4) only when expressed in an insect cell line; the proteins are also insensitive to in vitro phosphorylation by Fj or its mammalian homolog Fjx1. Nonetheless, the authors state:

“These data coincide with our findings that FJX1 does not phosphorylate Fat4 and imply that mammals encode an unidentified kinase that phosphorylates Fat proteins.”

But without evidence of phosphorylation this is more speculation than implication. Have the authors tried looking for phosphorylation after expression in additional mammalian cell lines?

Other notes:

Phosphorylated Fat and Ds for binding assays- I assume these were generated by incubation with Fj, but I could not find this in the methods.

Extended data Figure 8. What is in the “Ctl” lanes?

“Given that Fat and Dachous PCP is regulated by tension anisotropy”- This needs a reference.

Reviewer #2 (Remarks to the Author):

The authors report structures of a heterophilic complex of cadherin family proteins from the FAT and Dachous families. These enigmatic proteins, which are of enormous size, have not been well characterized before. Medina et al. report an anti-parallel EC1-EC4 heterocomplex structure, and expertly study the interaction chemistry, aided by calorimetry to reveal the energetic contributions of different parts and bits. This is a novel structure that will guide future studies into these developmentally crucial proteins. Lastly, they show that while the *Drosophila* Fat-Dachous interaction may be regulated by extracellular phosphorylation by the kinase Fj, they did not observe the same with the vertebrate homologs. While the authors cannot resolve the identity of the vertebrate FAT/Dachous kinase, or if it even exists, this is an important observation to be reported. Finally, the experiments appear to have been performed well. Small technical issues are mentioned below.

I only have one major point to make: The rigor of the paper builds on this statement, which is the first sentence of the Results: "The Fat4:Dchs1 binding interface has been mapped to the N-terminal EC1-4 domains of each protein." Can the authors please include the reference for this here? Confusingly, the second sentence of the Discussion is "We determined that, despite their giant ectodomains, Fat4 and

Dchs1 interact predominantly through their first four N-terminal EC domains in a manner that resembles protocadherin homodimers." This can be interpreted (and was interpreted by me) as a claim that the authors show the heterocomplex is formed by the EC1-4 domains, and the remaining of the giant proteins are not involved. That was not demonstrated here, however. The remaining claims of the manuscript are adequately supported.

Overall, the manuscript is easy to read and understand, but some parts can be improved in clarity and by being more precise (as mentioned above); see the following list for minor points, all of which can be addressed during revisions in a reasonable amount of time.

1. Authors state "Vertebrate Fat4 is the homolog of Drosophila Ft, and Dchs1 is *the* vertebrate homolog of Drosophila Ds". Do the authors simply mean Ft has the highest sequence sequence identity to Fat4 among vertebrate Fats? Or do they mean that there were two Fats in the common ancestor to both flies and vertebrates, and fly Ft and vertebrate Fat4 are the descendants of one of these ancestral Fats? If so, it would be great to see a reference for a phylogenetic study, or an argument based on, say, domain structure, or simply by sequence similarity. Same for Dachsous.

2. Authors state "Most of the Fat4 and Dchs1 EC domains are connected by calcium-binding motifs that coordinate up to three Ca²⁺ ions²⁷." It may make sense to also reference (28) here as that is a Fat-specific reference, and not a generic Calcium+Cadherin one.

3. Extended Data Figure 1: This figure could benefit in clarity from better labeling of images with regards to the protein expressed and the staining reagents used on the images.

4. The following statement is confusing: "The presence of an arginine at this position in Dchs1(EC4) may have biased the interaction to rely on the hydrophobic Fat4(EC4):Dchs1(EC1) region and corroborates our binding data".

Do the authors imply that that the difference in contribution to binding affinity between the two EC1-EC4 interactions is due to a difference in hydrophobic vs. polar interactions? The Dchs(EC4)-Fat(EC1) interaction is still looks reasonable as the R387 is satisfied with a negative charged E120 on the other side. It is indeed believable that the hydrophobic interaction is able to provide more binding energy. I am just not sure if this observation corroborates the binding data, or it provides a possible explanation for it (which is later supported by sequence conservation analysis, but not here).

5. "... the atomic contacts are all near the maximum allowed Van der Waal's distance (4-6 Å)."

Did the authors mean "acceptable as weak vdW contacts?" by "a maximum allowed vdW distance"? Should the "near" be replaced by "within" or not?

6. Authors state "the L379R mutation diminishes both the Dchs1-binding ability and the ability of Fat4 to form complexes." In addition to loss of the border complex, do the authors see general loss of adhesion between cells, or aggregation in classical cell aggregation experiments?

7. In Fig. 3C-E, there appears to be a lot of labeling of the membranes, likely as a result of over-expression. But also, the Fat4-L379R mutants appears to be more punctuate than wild-type. Could this be a result of misfolding and (consequently) mistargeting of the mutant into vesicles. Unless I misinterpreted it, Ext. Fig. 5 is not a surface staining experiment, and therefore cannot answer if the mutation caused folding/targeting defects. That can be an important control.

Also, are the border complexes measured in units of arbitrary fluorescence units-squared?

8. Ext. Fig. 2: Labeling chromatograms as "complex" when the claim is that no complex forms is confusing. Similarly, the authors can assist readers by stating what Streptavidin-488 labels in Ext. Fig. 4, at least in the legends.

9. In one of the validation reports provided for the lower-resolution structure, there is an odd comment about ligand identities: "Molecule 5 is CALCIUM ION (three-letter code: NAG) (formula: C8H15NO6)." These validation reports are not the versions endorsed for manuscript review by the PDB, and such an obvious mistake would have been corrected if that was the case. The authors should deposit and submit the version analyzed by PDB staff and labeled "For Manuscript Review" (which will still stay hidden until publication).

Also, while the low resolution of one of this structure do present challenges, additional effort to correct the high number of clashes may be beneficial. It may not be possible to improve the model, of course.

10. For the ITC data, I could not find where the authors report the whole set of fitting parameters (such as n [stoichiometry] and ΔH), and any goodness-of-fit, such as χ^2 . I would expect these to be included in a Table, and also placed in the main figures in reasonable detail whenever available (i.e. except in cases where binding is obviously too weak to quantitate accurately).

11. I was left wondering about the relevance may be Nicoludis et al. 2019 to this paper, where EC2-EC3 contacts were shown to be dynamic (i.e. liable to break) in MD simulations: I believe this may be where the authors also observed lack of contacts in their heterodimer. If relevant, the authors may like to discuss more of such literature from the Shapiro and Gaudet groups on the protocadherin structures.

Reviewer #3 (Remarks to the Author):

Please check the attached file.

Medina et al. report the crystal structure of human Fat4-Dchs1 complex in this manuscript. Previously, the N-terminal four extracellular cadherin (EC) domains of Fzt4 and Dchs1 were shown to be sufficient for Fat4-Dchs1 interactions, and the authors performed ITC experiments to confirm that the EC1-4 domains of Fzt4 are indispensable for Dchs1 binding but found that the EC4 of Dchs1 makes a minor contribution to the Fat4 interaction. Then, the authors solved the structures of Fat4(EC1-4):Dchs1(EC1-3) and Fat4(EC1-4):Dchs1(EC1-4) at 2.2 and 3.7 Å resolution, respectively. So far, many protocadherin and cadherin structures have been published but most of them form homodimers, except for the PCDH15:CDH23 heterodimer (EC1-2). Comparing this heterodimer structure with other homodimer structures revealed structural features stabilizing the heterotypic interaction between Fat4 and Dchs1. In addition, the authors performed cell-based experiments and computational analysis to complement structural data. This study does not cover the entire architecture of the giant proteins Fat4 and Dchs1 and does not describe the three dimensional organization of Fat4 and Dchs1 (clustering) in the confined intercellular spaces. Nevertheless, I think this study greatly improves the understanding of the interactions between minimal binding domains of Fat4 and Dchs1 at the molecular level. There are some issues which will need to be clarified before publishing will be considered.

Major issues;

1. The authors report the crystal structure of Fat4(EC1-4)-Dchs1(EC1-3) at 2.2 Å resolution, but the data collection statistics of the highest resolution shell (2.283-2.204) shown in extended data Table 1 looks pretty bad, such as 0.8 for “Mean I/sigma(I)”, 1.496 for “R_{merge}”, and 0.417 for “CC1/2”. Therefore, it is necessary to check whether this is actually a 2.2 Å resolution structure.
2. The authors say “We used the higher-resolution Fat4(EC1-4):Dchs1(EC1-3) structure for all subsequent analyses of interface contacts except for those involving EC4 of Dachsous1.” (lines 112-114), but the higher-resolution Fat4(EC1-4):Dchs1(EC1-3) structure is not shown anywhere in this manuscript. Please present the higher-resolution structure compared to the lower-resolution Fat4(EC1-4):Dchs1(EC1-4) structure, which could visualize limited interdomain flexibility as mentioned in lines 122-124.
3. Related to #2, the authors stated that the biggest structural difference between Fzt4:Dchs1 heteromer and other protocadherin homodimers is the large tilt of EC1 of Dchs1 towards Fat4. The authors should confirm that this structural feature is not an artifact due to crystal contacts. Also, the authors should show that both structures have similarly bent EC1 conformations. In addition, can the authors explain what causes such a large tilt in the EC1-EC2 linker of Dchs1 by structural analysis? For example, are there any sequence/structure variations in the calcium binding motif of the EC1-EC2 linker of Dchs1?

4. In Figure 6, the authors showed that phosphorylation in EC3 of Ft enhances the binding affinity to Ds, but did not describe how to prepare phosphorylated samples and confirm the exact phosphorylation site. If the authors performed an in vitro phosphorylation reaction, did the authors confirm single phosphorylation at the specified residue in the purified phospho-sample? Please provide detailed information about sample preparation and phospho-site identification.
5. In lines 254-258, the authors stated that phosphorylation strengthens Ft-Ds interactions. However, Brittle et al. and Hale et al. reported that phosphorylation of Ds by Fj reduces the binding affinity between Ft and Ds (doi.org/10.1016/j.cub.2010.03.056 and doi.org/10.7554/eLife.05789.001, respectively). Please explain the discrepancy.

Minor issues;

1. In page 2, references 3-5 are introduced first, followed by references 1,2. Please list them in order starting with reference 1.
2. In Figures 1c and 6b, and extended data figure 3, representative ITC thermograms are presented but thermodynamic parameters are missing. Please include thermodynamic parameters for all ITC data (supplementary table would be good).
In addition, the thermogram measured from phosphorylated Ft and non-phosphorylated Ds shows that the interaction is endothermic reaction and the molar ratio is close to 0.5, not 1.0. Do the authors have any comments on this?
3. In extended data Figure 2, it is strange that the elution volume of Fat4(EC1-4) and Dchs1(EC1-4) in figure (a) (~13.5 ml) does not match well in figures (b) Dchs1(EC1-4) (~13 ml) and (c) Fat4(EC1-4) (~13 ml). Any comments?
4. In extended data figure 8(b), not only FJX1 but also all (?) other proteins seem to have His tag. Please provide the sample information (whether they have His tag or not) in the figure legend.
5. For ITC experiment and complex formation, the authors purified monomers of Fat4 and Dchs1, but in the cell-based binding test (extended data figure 4), the authors used recombinant Fat/Fat4^{L379R} tetramers. Please provide the method how to prepare these recombinant tetramers. In extended data figure 4, please consistently write L379R as superscript in "Fat4L379R".

6. Alphafold model of Ft(EC1-4):Ds(EC1-4) doesn't seem to have Ca^{2+} ions. If so, the authors should mention that calcium ions are not included in the modeling.

Did the authors try to run an energy minimization of the model including Ca^{2+} ions, because Ft and Ds have calcium ions?

RESPONSE TO REVIEWERS

Manuscript ID: NCOMMS-22-35467-T

We thank the reviewers for their helpful comments to improve manuscript quality and clarity. We have performed additional experiments and edited multiple figures and the text to address the below comments. Experimentally, we have (a) performed ITC experiments containing both phosphorylated Fat and Dachsoous, (b) included confocal images of HEK293 cells expressing wild-type and L379R Fat4 to show no effect on surface localization, and (c) reprocessed the high-resolution dataset to 2.3 angstrom to improve statistical parameters. Specific changes to the manuscript are highlighted under each comment below. We hope these modifications will satisfy the reviewers and make the manuscript suitable for publication.

REVIEWER COMMENTS

Reviewer #1 (Remarks to the Author):

- 1) While phosphorylated and non-phosphorylated Dachsoous(EC1-4) bound unphosphorylated Fat(EC1-4) too weakly to detect a difference, I wonder why the authors did not test whether phosphorylation of Dachsoous(EC1-4) changed its affinity for phosphorylated Fat(EC1-4)?

We agree that it would be useful to test the binding of phosphorylated Ft with phosphorylated Ds. We performed the experiments combining both phosphorylated Ft(EC1-4) and phosphorylated Ds(EC1-4) and have determined a binding affinity of $12.5 \pm 3.79 \mu\text{M}$, placing the combination between phosphorylated Ft alone and the wild-type interaction. We have updated Figure 6 and the manuscript to include this new data.

- 2) Nor did the authors discuss possible effects on the binding structure that would be predicted from phosphorylation of Dachsoous EC3, and why this might lead to weaker binding.

We have rewritten the following sentences in the section **Phosphoregulation of Fat-Ds interactions in *Drosophila*** and the new section now reads as below (changes in **bold**):

“The Ft phosphate was positioned within 5 Å of the positively-charged guanidinium group of Ds-Arg177, and the Ds phosphate was approximately 6.3 Å from the hydroxyl group of Ft-Thr210 (Figure 6A). **Further, the majority of residues modeled in the Ft(EC2) β4-β5 loop are negatively charged.** This positioning suggests that phosphorylation of Ft can strengthen Ft:Ds interactions by introducing polar contacts across the binding interface, likely by generating a salt bridge with Arg177 of Ds. **Meanwhile, phosphorylation of Ds can introduce repulsive forces into the Ft(EC2):Ds(EC3) interface which may destabilize the interaction.** Although the predicted interactions are not close contacts ($\geq 5 \text{ Å}$), we anticipate that minor errors in the model and fluctuations in rotamer or domain orientations could account for this disparity.”

We also updated the PhosphoDachsoous panel in **Figure 6A** to display all the negatively charged residues in the loop mentioned.

There is also one prior result that might be worth discussing, as it suggests that phosphorylation of domains other than the EC3 might modify binding: Simon et al. claimed that Fj coexpression

still increased the affinity of Fat for Dachshous even when the Fat lacked its EC3 phosphorylation site, albeit not as strongly as with wild type Fat (see their Fig. S3).

3) Could the authors comment briefly on the possible roles of the other EC Fj targets?

This is an interesting question. Based on the SPR studies by Tsukasaki, et al. (PNAS, 2014), which showed that binding of the full Fat4/Dchs1 ECDs was equivalent to EC1-4 binding, we think it is unlikely that phosphorylation of other putative EC targets would directly impact the binding interface. In Loza, et al. (eLife, 2014), Fat4 and Dchs1 were found to be clustered at cell-cell contacts, and this clustering may be enhanced or inhibited by modification of these other sites. It is therefore possible that phosphorylation affects oligomerization, which would in turn affect binding avidity between cells.

Based on our finding that the established Fj motif is insufficient to predict phosphorylation, it is also interesting to consider whether Four-jointed may be capable of phosphorylating EC repeats beyond Fat/Dachshous. We have added some discussion pointing out that Flamingo (and the human ortholog CELSR) contain the phosphorylation motif as well and is conserved between flies and humans.

More difficult is drawing parallels with Fat4 and Dchs1, as the authors have not detected any phosphorylation of Fat4(EC1-4), and weak phosphorylation of Dchs1(EC1-4) only when expressed in an insect cell line; the proteins are also insensitive to in vitro phosphorylation by Fj or its mammalian homolog Fjx1. Nonetheless, the authors state:

“These data coincide with our findings that FJX1 does not phosphorylate Fat4 and imply that mammals encode an unidentified kinase that phosphorylates Fat proteins.”

4) But without evidence of phosphorylation this is more speculation than implication. Have the authors tried looking for phosphorylation after expression in additional mammalian cell lines?

We have not looked for phosphorylation after expression in additional mammalian cell lines. The subject of whether Fat4 is phosphorylated at all in mammals is an interesting question, but we believe this is outside the scope of our study. We have edited the passage to acknowledge that we do not know whether the Fat4 ectodomain is phosphorylated in mammals, and to mention that Fjx1 does not phosphorylate Fat1 or Fat4.

Other notes:

5) Phosphorylated Fat and Ds for binding assays- I assume these were generated by incubation with Fj, but I could not find this in the methods.

The method for phosphorylation can be found under the heading **Phosphorylation studies** subheading **In-vitro Phosphorylation**.

6) Extended data Figure 8. What is in the “Ctrls” lanes?

The control lanes are the same unphosphorylated and phosphorylated protein ladders as used in Figures 5C and 5D. We have updated the figure legends for Figures 5C and 5D, and Extended Data Figure 8 to reflect this.

- 7) "Given that Fat and Dachsous PCP is regulated by tension anisotropy"- This needs a reference.

We referenced Bosveld, et al. (2012) here.

Reviewer #2 (Remarks to the Author):

I only have one major point to make:

- 1) The rigor of the paper builds on this statement, which is the first sentence of the Results: "The Fat4:Dchs1 binding interface has been mapped to the N-terminal EC1-4 domains of each protein." Can the authors please include the reference for this here?

We apologize for this oversight. We have now referenced the PNAS article from Tsukasaki, et al. here. We also changed the first sentence of our results section to more clearly state was shown in the referenced paper.

- 2) Confusingly, the second sentence of the Discussion is "We determined that, despite their giant ectodomains, Fat4 and Dchs1 interact predominantly through their first four N-terminal EC domains in a manner that resembles protocadherin homodimers." This can be interpreted (and was interpreted by me) as a claim that the authors show the heterocomplex is formed by the EC1-4 domains, and the remaining of the giant proteins are not involved. That was not demonstrated here, however. The remaining claims of the manuscript are adequately supported.

We agree that this was confusing and revised the discussion to state, "We determined that EC1-4 of Fat4 and Dchs1 interact in a manner that resembles protocadherin homodimers".

Overall, the manuscript is easy to read and understand, but some parts can be improved in clarity and by being more precise (as mentioned above); see the following list for minor points, all of which can be addressed during revisions in a reasonable amount of time.

- 3) Authors state "Vertebrate Fat4 is the homolog of Drosophila Ft, and Dchs1 is *the* vertebrate homolog of Drosophila Ds". Do the authors simply mean Ft has the highest sequence identity to Fat4 among vertebrate Fats? Or do they mean that there were two Fats in the common ancestor to both flies and vertebrates, and fly Ft and vertebrate Fat4 are the descendants of one of these ancestral Fats? If so, it would be great to see a reference for a phylogenetic study, or an argument based on, say, domain structure, or simply by sequence similarity. Same for Dachsous.

We have referenced Rock, et al. (2005) here, as this appears to be the earliest paper to determine conservation between all fly and vertebrate Fat/Dachsous proteins. We also edited this passage to more clearly state the relationship between Drosophila Fat/Ds to human Fat4/Dchs1.

- 4) Authors state "Most of the Fat4 and Dchs1 EC domains are connected by calcium-binding motifs that coordinate up to three Ca²⁺ ions²⁷." It may make sense to also reference (28) here as that is a Fat-specific reference, and not a generic Calcium+Cadherin one.

We have included the suggested reference.

- 5) Extended Data Figure 1: This figure could benefit in clarity from better labeling of images with regards to the protein expressed and the staining reagents used on the images.

We have reformatted the figure to better clarify the protein overexpressed within the image and the staining reagent used.

- 6) The following statement is confusing: "The presence of an arginine at this position in Dchs1(EC4) may have biased the interaction to rely on the hydrophobic Fat4(EC4):Dchs1(EC1) region and corroborates our binding data".

Do the authors imply that that the difference in contribution to binding affinity between the two EC1-EC4 interactions is due to a difference in hydrophobic vs. polar interactions? The Dchs(EC4)-Fat(EC1) interaction is still looks reasonable as the R387 is satisfied with a negative charged E120 on the other side. It is indeed believable that the hydrophobic interaction is able to provide more binding energy. I am just not sure if this observation corroborates the binding data, or it provides a possible explanation for it (which is later supported by sequence conservation analysis, but not here).

We have revised the passage to state how the central arginine creates less hydrophobic interface and speculate that this might lead to a reduced contribution to overall binding energy.

- 7) "... the atomic contacts are all near the maximum allowed Van der Waal's distance (4-6 Å)."

Did the authors mean "acceptable as weak vdW contacts?" by "a maximum allowed vdW distance"? Should the "near" be replaced by "within" or not?

We have edited the passage to reflect our referring to weak vdW contacts.

- 8) Authors state "the L379R mutation diminishes both the Dchs1-binding ability and the ability of Fat4 to form complexes." In addition to loss of the border complex, do the authors see general loss of adhesion between cells, or aggregation in classical cell aggregation experiments?

We thank the reviewer for raising this point. We indeed do not test the effect of the mutation in Fat4 on cell adhesion, but rather focus on the boundary accumulation, which is a more direct measure of the interaction between boundary complexes. We believe the effect on cell-cell adhesion, while interesting, could be a downstream consequence of the reduced binding at the boundary, but is beyond the scope of this work.

- 9) In Fig. 3C-E, there appears to be a lot of labeling of the membranes, likely as a result of over-expression. But also, the Fat4-L379R mutants appears to be more punctuate than wild-type. Could thus be a result of misfolding and (consequently) mistargeting of the mutant into vesicles. Unless I misinterpreted it, Ext. Fig. 5 is not a surface staining experiment, and therefore cannot answer if the mutation caused folding/targeting defects. That can be an important control.

The reviewer raises an important point regarding the localization of the mutated Fat4-L379R. We have now added an additional panel to Extended Data Figure 6 showing representative images of Fat4-WT and Fat4-L379R mono-cultures. These show that both variants primarily

localize to the membrane, and that there is no clear difference between their subcellular localization. In general, looking at all our images both in monocultures and in co-cultures we do not see any apparent differences in localization between the WT and mutant variant. We have now corrected the text to address that:

“This reduction was not due to differences in expression or subcellular localization between Fat4^{L379R} and wildtype Fat4 (Extended Data Figure 6).”

Also, are the border complexes measured in units of arbitrary fluorescence units-squared?

This is a good point. The y axis in Fig.3e is fluorescence in red times green fluorescence. The units are still arbitrary units (you do not usually square AU) but the label is now changed to (Red fl. X Green fl. [AU]) so that it is clear.

- 10)** Ext. Fig. 2: Labeling chromatograms as "complex" when the claim is that no complex forms is confusing. Similarly, the authors can assist readers by stating what Streptavidin-488 labels in Ext. Fig. 4, at least in the legends.

We have updated the chromatogram labels to describe the mixture of Fat4 and Dchs1 proteins and not simply “complex”. We also updated the figure legend for Extended Data Figure 5 similarly to the changes in Extended Data Figure 1 to better describe the cells being stained and the reagent used to stain them.

- 11)** In one of the validation reports provided for the lower-resolution structure, there is an odd comment about ligand identities: "Molecule 5 is CALCIUM ION (three-letter code: NAG) (formula: C8H15NO6)." These validation reports are not the versions endorsed for manuscript review by the PDB, and such an obvious mistake would have been corrected if that was the case. The authors should deposit and submit the version analyzed by PDB staff and labeled "For Manuscript Review" (which will still stay hidden until publication).

Also, while the low resolution of one of this structure do present challenges, additional effort to correct the high number of clashes may be beneficial. It may not be possible to improve the model, of course.

We have included the new validation report with this resubmission. In reference to the clashes observed in the low-resolution structure, it has been difficult resolving them without it impacting the Rfree for the overall model so we did our best to balance the two parameters.

- 12)** For the ITC data, I could not find where the authors report the whole set of fitting parameters (such as n [stoichiometry] and deltaH), and any goodness-of-fit, such as chi². I would expect these to be included in a Table, and also placed in the main figures in reasonable detail whenever available (i.e. except in cases where binding is obviously too weak to quantitate accurately).

We have included fitting parameters for all displayed thermograms in a supplementary table. We have also edited the manuscript to refer to the table whenever discussing an ITC result.

- 13) I was left wondering about the relevance may be Nicoludis et al. 2019 to this paper, where EC2-EC3 contacts were shown to be dynamic (i.e. liable to break) in MD simulations: I believe this may be where the authors also observed lack of contacts in their heterodimer. If relevant, the authors may like to discuss more of such literature from the Shapiro and Gaudet groups on the protocadherin structures.

We thank the reviewer for calling our attention to Nicoludis, et al. 2019 and agree that these insights are relevant to our study. We added this reference and the following sentence, “For example, EC2-EC3 contacts in protocadherins have been found to “break” in molecular dynamics simulations, and a similar disengagement in Fat4-Dchs1 could push the Fj-modified phosphoserines into contact with interface residues.” Together with our new ITC experiment indicating phosphoDs has reduced binding affinity for phosphoFat, we can imagine situations where modification of Fat or Ds can tune the overall binding affinity according to the needs of cells/tissues. We have updated the Discussion to reflect this thinking and have referenced accordingly.

Reviewer 3 comments

Major issues:

- 1) The authors report the crystal structure of Fat4(EC1-4)-Dchs1(EC1-3) at 2.2 Å resolution, but the data collection statistics of the highest resolution shell (2.283-2.204) shown in extended data Table 1 looks pretty bad, such as 0.8 for “Mean I/sigma(I)”, 1.496 for “Rmerge”, and 0.417 for “CC1/2”. Therefore, it is necessary to check whether this is actually a 2.2 Å resolution structure.

Upon truncation of the dataset to 2.3 angstrom resolution, the data in the highest resolution shell was: 1.38 I/sigma, 0.95 Rmerge, 0.582 CC1/2. Rfree also decreased (23.5% from 24.7%). Based these changes, we decided to truncate the data at 2.3 angstroms rather than 2.2. We thank the reviewer for this observation and have updated Extended Data Table 1 to include the new statistics.

- 2) The authors say “We used the higher-resolution Fat4(EC1-4):Dchs1(EC1-3) structure for all subsequent analyses of interface contacts except for those involving EC4 of Dachsous1.” (lines 112-114), but the higher-resolution Fat4(EC1-4):Dchs1(EC1-3) structure is not shown anywhere in this manuscript. Please present the higher-resolution structure compared to the lower-resolution Fat4(EC1-4):Dchs1(EC1-4) structure, which could visualize limited interdomain flexibility as mentioned in lines 122-124.

We have included a new supplementary figure (**Extended Data Figure 3**) that displays the high-resolution structure similarly to how the low-resolution structure is displayed in Figure 2 and superimposes the low- and high-resolution structures globally to show the limited interdomain flexibility. We have also adjusted the rest of the extended data figure numbering in accommodation.

- 3) Related to #2, the authors stated that the biggest structural difference between Fzt4:Dchs1 heteromer and other protocadherin homodimers is the large tilt of EC1 of Dchs1 towards Fat4. The authors should confirm that this structural feature is not an artifact due to crystal contacts. Also, the authors should show that both structures have similarly bent EC1 conformations. In addition, can the authors explain what causes such a large tilt in the EC1-EC2 linker of Dchs1 by structural analysis? For example, are there

any sequence/structure variations in the calcium binding motif of the EC1-EC2 linker of Dchs1?

We have created a new supplementary figure (**Extended Data Figure 8A**) aligning the protocadherin structures being compared with Dchs1(EC2) and the characteristic “tilt” of Dchs1(EC1) can be seen in the high- and low-resolution models. Further, both structures crystallized in different space groups, and while in the Fat4(EC1-4):Dchs1(EC1-3) structure Dchs1(EC1) makes putative contact with only one symmetry chain the same domain makes contact with up to four different symmetry chains in the Fat4(EC1-4):Dchs1(EC1-4) structure.

There does not appear to be any structural variance within the calcium-binding motif of the EC1-EC2 linker. However, the β 5- β 6 loop in nearly every member of each protocadherin family contains a disulfide-linked stretch of 7 residues (including the cysteines) that appears to promote interdomain contacts with the β 2- β 3 loop of EC2. This is absent in both Fat4 and Dchs1 and it makes the loop region 4 residues shorter in Fat4 and 6 residues shorter in Dchs1. This truncation may permit the EC1-EC2 tilt due to a larger degree of freedom. We have made a supplementary figure (**Extended Data Figure 8B**) of a sequence alignment for the specific area to highlight the truncation and now discuss the structural feature within the manuscript.

- 4) In Figure 6, the authors showed that phosphorylation in EC3 of Ft enhances the binding affinity to Ds, but did not describe how to prepare phosphorylated samples and confirm the exact phosphorylation site. If the authors performed an in vitro phosphorylation reaction, did the authors confirm single phosphorylation at the specified residue in the purified phospho-sample? Please provide detailed information about sample preparation and phospho-site identification.

The protein used during the ITC experiments is the same as that used in the phosphogels seen in Figure 5, and so the phospho-site location is the same as seen in the gels. For the ITC experiments a scaled-up version of the phosphorylation reaction described under heading **Phosphorylation studies** subheading **In vitro Phosphorylation** in the Materials and Methods was used, followed by polishing using size exclusion chromatography. We have included a sentence in the **ITC binding studies** section of the Materials and Methods to indicate this.

- 5) In lines 254-258, the authors stated that phosphorylation strengthens Ft-Ds interactions. However, Brittle et al. and Hale et al. reported that phosphorylation of Ds by Fj reduces the binding affinity between Ft and Ds (doi.org/10.1016/j.cub.2010.03.056 and doi.org/10.7554/eLife.05789.001, respectively). Please explain the discrepancy.

Reviewer 1 referred similarly to this section in regards to a lack of structural analysis concerning how phosphorylated Ds may impact Ft:Ds interactions and we realize this passage may sound like we are discussing phosphorylation generally when we were only referring to phosphorylation of Fat. We have rewritten the passage as below:

“The Ft phosphate was positioned within 5 Å of the positively-charged guanidinium group of Ds-Arg177, and the Ds phosphate was approximately 6.3 Å from the hydroxyl group of Ft-Thr210 (**Figure 6A**). **Further, the majority of residues modeled in the Ft(EC2) β 4- β 5 loop are negatively charged.** This positioning suggests that phosphorylation of Ft can strengthen Ft:Ds interactions by introducing polar contacts across the binding interface, likely by generating a salt bridge with Arg177 of Ds. **Meanwhile, phosphorylation of Ds can introduce repulsive forces into the Ft(EC2)-Ds(EC3) interface which may destabilize the interaction.** Although the

predicted interactions are not close contacts ($\geq 5 \text{ \AA}$), we anticipate that minor errors in the model and fluctuations in rotamer or domain orientations could account for this disparity.”

We also updated the PhosphoDachsous panel in **Figure 6A** to display all the negatively charged residues in the loop mentioned.

Minor issues;

- 6) In page 2, references 3-5 are introduced first, followed by references 1,2. Please list them in order starting with reference 1.

References have been updated to reflect proper numbering.

- 7) In Figures 1c and 6b, and extended data figure 3, representative ITC thermograms are presented but thermodynamic parameters are missing. Please include thermodynamic parameters for all ITC data (supplementary table would be good). In addition, the thermogram measured from phosphorylated Ft and non-phosphorylated Ds shows that the interaction is endothermic reaction and the molar ratio is close to 0.5, not 1.0. Do the authors have any comments on this?

Despite matching the experimental conditions for the human Fat4:Dchs1 experiments we were surprised to see an endothermic reaction with fly Fat and Dachsous. For the thermograms displayed the Origin software determined a N-value of ~ 0.75 for both the phosFat:WTDs and phosFt:phosDs interactions. A partial contributor to the lower-than-expected N-value could be due to variability in determining concentration (neither protein contains Tryptophans) coupled with the lower binding affinities observed. We have included a supplementary table (Extended Data Table 6) containing the thermodynamic parameters for the thermograms displayed in Figures 1C and 6B, and Extended Data Figure 3. The manuscript has been updated to refer to the supplementary table in appropriate sections.

- 8) In extended data Figure 2, it is strange that the elution volume of Fat4(EC1-4) and Dchs1(EC1-4) in figure (a) ($\sim 13.5 \text{ ml}$) does not match well in figures (b) Dchs1(EC1-4) ($\sim 13 \text{ ml}$) and (c) Fat4(EC1-4) ($\sim 13 \text{ ml}$). Any comments?

The chromatograms are derived from preparative columns and were not performed at the same time so some variability in retention volume would be expected due to drift commonly seen with instrument wear and tear.

- 9) In extended data figure 8(b), not only FJX1 but also all (?) other proteins seem to have His tag. Please provide the sample information (whether they have His tag or not) in the figure legend.

We have updated the figure legend to reflect that all the proteins contain His-tags.

- 10) For ITC experiment and complex formation, the authors purified monomers of Fat4 and Dchs1, but in the cell-based binding test (extended data figure 4), the authors used recombinant Fat/Fat4L379R tetramers. Please provide the method how to prepare these recombinant tetramers. In extended data figure 4, please consistently write L379R as superscript in “Fat4L379R”.

We have revised the subsection **Binding tests** within the section **Confocal Microscopy** within the Materials and Methods to emphasize C-terminally biotinylated Fat4/Dchs1 were incubated with Streptavidin-AF488/AF647 prior to cell staining. We have also added the superscript annotations.

- 11)** Alphafold model of Ft(EC1-4):Ds(EC1-4) doesn't seem to have Ca²⁺ ions. If so, the authors should mention that calcium ions are not included in the modeling. Did the authors try to run an energy minimization of the model including Ca²⁺ ions, because Ft and Ds have calcium ions?

We have updated the Figure 6 legend to indicate the Alphafold model does not contain Calcium ions, and acknowledge it in the text. However, structural alignment of pairs of EC domains from the Alphafold model strongly resembled those from the experimentally determined Fat4 and Dchs1 structures, so we do not believe the lack of calcium ions substantially altered the relative domain positions (see below). We also included commentary indicating that the lack of calcium ions in the model may contribute slight effects to the positions of the Fj-phosphorylated residues.

Interface	RMSD between Fat4 and Ft model (Å)	RMSD between Dchs1 and Ds model (Å)
EC1-2	1.4	1.6
EC2-3	1.1	1.7
EC3-4	1.7	2.4

REVIEWERS' COMMENTS

Reviewer #1 (Remarks to the Author):

The manuscript is improved, and I thank the authors for the valuable additional data and discussion. I have a few relatively minor points which I would like the authors to consider, but I do not need to review these before publication.

1) Fat 4 vs Fat 1,2,and 3- First, a point brought up by Reviewer 2, who asks:

“Authors state "Vertebrate Fat4 is the homolog of Drosophila Ft, and Dchs1 is *the* vertebrate homolog of Drosophila Ds". Do the authors simply mean Ft has the highest sequence identity to Fat4 among vertebrate Fats? Or do they mean that there were two Fats in the common ancestor to both flies and vertebrates, and fly Ft and vertebrate Fat4 are the descendants of one of these ancestral Fats? If so, it would be great to see a reference for a phylogenetic study, or an argument based on, say, domain structure, or simply by sequence similarity. Same for Dachsous.”

In reply, the authors seem to be going for the greater sequence identity answer, and have changed their text to read:

“In Drosophila, there are two Fat proteins (Ft and Ft-like) and one Dachsous protein 48 (Ds). In vertebrates, there are four Fat proteins (Fat1-4) and two Dachsous proteins (Dchs1-2). 49 Vertebrate Fat4 and Dchs1 have the highest sequence identity to Drosophila Ft and Ds, 50 respectively, and Fat4 has been shown to directly interact with Dchs110”

But analyses indicate that there are in fact two distinct Fats in the common ancestor of Drosophila and vertebrates, Fat and Fat-like. See Castillejo et al. 2004, which first identified Fat-like in flies and noted its greater similarity to mammalian Fat1,2 and 3 than to Drosophila Fat (admittedly written before the first publication on vertebrate Fat4/Fat-j that same year). Rock et al. make a brief analysis of similarities, and Hulpiau and Van Roy 2009 make it quite clear that there are two distinct Fat and Fat-like families. In fact,

the two families appear to predate bilateria, as they can now be found in sequences in the Cnidarian *Nematostella*.

Family membership is based not only on sequence similarities in the ECD and ICD, but also domain structure: Fat proteins have two and Fat-like proteins have one laminin G-like domain. Binding to Ds proteins has so far only been found for Fat family members.

The changed text also makes it sound like reference 10 contains Fat4: Dchs1 binding data, which it does not.

Castillejo-Lopez et al., 2004. The fat-like Gene of *Drosophila* Is the True Orthologue of Vertebrate Fat Cadherins and Is Involved in the Formation of Tubular Organs.

Hulpiau and van Roy (2009). Molecular evolution of the cadherin superfamily.

2) Conservation

“Fat and Dachshous homologs are encoded

224 by organisms ranging from hydras to humans^{7,39}. We performed a conservation analysis to
225 investigate whether the binding mode observed in the human Fat4:Dchs1 structure is broadly
226 representative of Fat-Dachshous interactions (Extended Data Table 5). The residues forming the
227 Fat4(EC1):Dchs1(EC4), Fat4(EC2):Dchs1(EC3), and Fat4(EC4):Dchs1(EC1) interfaces are
228 mostly conserved across 18 different species. The L379 residue in Fat4(EC4), which is critical for
229 Dchs1 interactions (Figure 3B) was also conserved and only varies between leucine and valine.
230 Within Dchs1(EC4), the Arg387 residue central to the Fat4-binding interface is highly conserved
231 across all species sampled. On the other hand, opposing residues in the minimally-contacting
232 Fat4(EC3) and Dchs1(EC2) domains are poorly conserved, which is consistent with a lesser role
233 in Fat4:Dchs1 binding (Figure 2A, Figure 5A). Taken together, these findings suggest that Fat
234 and Dachshous interact via a similar binding mode in various metazoan organisms.”

First, it is hard to tell from this paragraph which figure shows the conservation data. 5A?

Second, it implies a rather broad sampling of taxa “from hydra to humans”. But except for *Drosophila*, all the species used are vertebrate, and most mammalian. And so far it looks like hydra only have a Fat-like, not a Fat.

On a similar subject, what species were used for the following statement, and what species have lost the phosphorylation site in Fat4?

“Alignment of the

236 EC3 domains revealed that the Fj phosphorylation motif (D-X-N-D-[X]7-S/T) was fully conserved
237 in Dachous homologs and mostly conserved in Fat homologs, including human Fat4 (Figure
238 5B).:

3) Ds-independent activity

“Fat proteins are multifunctional and appear to have distinct Dachous-dependent and
343 ligand-independent functions. For example, Fat-Dachous interactions regulate PCP, the Fat4
344 ICD modulates Hippo signaling by engaging the scaffold protein Angiomotin-like 1 (Amotl1), and
345 Fat4 inhibits RET signaling to guide kidney development. Additionally, nearly all cancer
346 associated Fat4 and Dchs1 mutations in the COSMIC database are located outside of the binding
347 interface, suggesting that the tumor suppressor role of Fat and Dachous may not require their
348 direct interactions.”

First, in most situations Fat-Ds binding is thought to modulate signaling through the Fat or Ds ICDs. In that context, finding critical mutations in other parts of the Fat4 or Dchs1, as has been done for fly Fat, is not surprising, and does not rule out a critical role for binding. As for the Amotl1 and RET studies, were the effects ever shown to be independent of Dchs proteins? Better might be the fly studies, where simultaneous removal of both Fat and Ds has stronger effects on growth than removal of Ds. But even that result is complicated by the possibility of opposing activities of the Fat and Ds ICDs.

Second, please provide and state references here. Zhang et al for Fat4 and RET was not mentioned previously and is not in the reference list.

4) Null binding

“Through our mutational studies, we isolated and characterized null-binding 349 L379R and R297A mutants in Fat4 and Dchs1, respectively.”

The term “null-binding mutants” implies the complete absence of binding, while I think the authors can only claim a strong reduction, especially in the context of full-length proteins, where the sole analysis is reduced colocalization at cell interfaces.

The reduced binding of mutant Fat4 EC1-4 to Dchs1 on cells in Extended Fig 5 is not very convincing, as the cells shown also have, in contrast to the controls, almost no Dchs1 on the cell surface.

Reviewer #2 (Remarks to the Author):

The authors have comprehensively responded to my comments, as well as those of the other reviewers. All my major and minor points have been adequately addressed. I have two **very** minor points with the revisions to mention:

1. In response to reviews, the authors cut the resolution of one of their crystal structures, effectively removing data. There is a well-documented "perverse incentive" (PMID: 26209821) among scientists to cut their data, which usually results in decreases in Rmerge and in R/Rfree, to satisfy reviewers. High Rmerge and Rfree values in the highest resolution bin do not serve as a valid reason to remove valid data when CC1/2 values indicate otherwise. However, I respect the choice of the author's response to reviewer's comments.

2. Authors responded that updated validation reports are provided. I could not see these reports. Likely a problem with the manuscript submission system.

Reviewer #3 (Remarks to the Author):

The revised manuscript by Medina et al. entitled “Structure of the planar cell polarity cadherins Fat4 and Dachsous1”, and the detailed rebuttal letter provided missing details. All issues raised in my report have been satisfactorily addressed and the manuscript has been improved. I would like to point out one minor thing in extended data table 6. Usually, ΔG values are also given in thermodynamic table along with ΔH and ΔS (although those can be easily calculated by ΔH and ΔS), and appropriate units should be provided with ΔH and ΔS , such as cal/mol and cal/mol K.

We thank the reviewers for their feedback and have addressed the points discussed.

Reviewer's Comments:

Reviewer #1 (Remarks to the Author)

The manuscript is improved, and I thank the authors for the valuable additional data and discussion. I have a few relatively minor points which I would like the authors to consider, but I do not need to review these before publication.

1) Fat 4 vs Fat 1,2,and 3- First, a point brought up by Reviewer 2, who asks:

“Authors state "Vertebrate Fat4 is the homolog of Drosophila Ft, and Dchs1 is *the* vertebrate homolog of Drosophila Ds". Do the authors simply mean Ft has the highest sequence identity to Fat4 among vertebrate Fats? Or do they mean that there were two Fats in the common ancestor to both flies and vertebrates, and fly Ft and vertebrate Fat4 are the descendants of one of these ancestral Fats? If so, it would be great to see a reference for a phylogenetic study, or an argument based on, say, domain structure, or simply by sequence similarity. Same for Dachsous.”

In reply, the authors seem to be going for the greater sequence identity answer, and have changed their text to read:

“In Drosophila, there are two Fat proteins (Ft and Ft-like) and one Dachsous protein 48 (Ds). In vertebrates, there are four Fat proteins (Fat1-4) and two Dachsous proteins (Dchs1-2).
49 Vertebrate Fat4 and Dchs1 have the highest sequence identity to Drosophila Ft and Ds,
50 respectively, and Fat4 has been shown to directly interact with Dchs110”

But analyses indicate that there are in fact two distinct Fats in the common ancestor of Drosophila and vertebrates, Fat and Fat-like. See Castillejo et al. 2004, which first identified Fat-like in flies and noted its greater similarity to mammalian Fat1,2 and 3 than to Drosophila Fat (admittedly written before the first publication on vertebrate Fat4/Fat-j that same year). Rock et al. make a brief analysis of similarities, and Hulpiau and Van Roy 2009 make it quite clear that there are two distinct Fat and Fat-like families. In fact, the two families appear to predate bilateria, as they can now be found in sequences in the Cnidarian Nematostella.

Family membership is based not only on sequence similarities in the ECD and ICD, but also domain structure: Fat proteins have two and Fat-like proteins have one laminin G-like domain. Binding to Ds proteins has so far only been found for Fat family members.

The changed text also makes it sound like reference 10 contains Fat4: Dchs1 binding data, which it does not.

Castillejo-Lopez et al., 2004. The fat-like Gene of Drosophila Is the True Orthologue of Vertebrate Fat Cadherins and Is Involved in the Formation of Tubular Organs.

Hulpiau and van Roy (2009). Molecular evolution of the cadherin superfamily.

We reworded this passage (see below) and replaced the Brittle reference with the Hulpiou reference:

“In *Drosophila*, there are two Fat proteins (Ft and Ft-like) and one Dachous protein (Ds). In vertebrates, there are four Fat proteins (Fat1-4) and two Dachous proteins (Dchs1-2). Fat1-3 are homologs of Ft-like, Fat4 is the homolog of Ft, and Dchs1 is the homolog of Ds¹⁰.”

2) Conservation

“Fat and Dachous homologs are encoded 224 by organisms ranging from hydras to humans^{7,39}. We performed a conservation analysis to 225 investigate whether the binding mode observed in the human Fat4:Dchs1 structure is broadly 226 representative of Fat-Dachous interactions (Extended Data Table 5). The residues forming the 227 Fat4(EC1):Dchs1(EC4), Fat4(EC2):Dchs1(EC3), and Fat4(EC4):Dchs1(EC1) interfaces are 228 mostly conserved across 18 different species. The L379 residue in Fat4(EC4), which is critical for 229 Dchs1 interactions (Figure 3B) was also conserved and only varies between leucine and valine. 230 Within Dchs1(EC4), the Arg387 residue central to the Fat4-binding interface is highly conserved 231 across all species sampled. On the other hand, opposing residues in the minimally-contacting 232 Fat4(EC3) and Dchs1(EC2) domains are poorly conserved, which is consistent with a lesser role 233 in Fat4:Dchs1 binding (Figure 2A, Figure 5A). Taken together, these findings suggest that Fat 234 and Dachous interact via a similar binding mode in various metazoan organisms.”

First, it is hard to tell from this paragraph which figure shows the conservation data. 5A?

We added an additional sentence that explicitly states binding residues were then painted according to their level of conservation after the sentence introducing the conservation analysis.

Second, it implies a rather broad sampling of taxa “from hydra to humans”. But except for *Drosophila*, all the species used are vertebrate, and most mammalian. And so far it looks like hydra only have a Fat-like, not a Fat.

The reviewer makes a good point here. We changed “hydra” to “Cnidarians” to clarify Fat can be found in Cnidaria but not necessarily in all members of Cnidaria (which includes *Hydra*). We also clarified the conservation analysis was done using the listed species in relevant sections.

On a similar subject, what species were used for the following statement, and what species have lost the phosphorylation site in Fat4?

“Alignment of the 236 EC3 domains revealed that the Fj phosphorylation motif (D-X-N-D-[X]7-S/T) was fully conserved 237 in Dachous homologs and mostly conserved in Fat homologs, including human Fat4 (Figure 238 5B).:”

We added a couple of words clarifying the EC3 domains were from the sample organism list used for the conservation analysis.

3) Ds-independent activity

“Fat proteins are multifunctional and appear to have distinct Dachsous-dependent and
343 ligand-independent functions. For example, Fat-Dachsous interactions regulate PCP, the
Fat4
344 ICD modulates Hippo signaling by engaging the scaffold protein Angiomotin-like 1 (Amotl1),
and
345 Fat4 inhibits RET signaling to guide kidney development. Additionally, nearly all cancer
346 associated Fat4 and Dchs1 mutations in the COSMIC database are located outside of the
binding
347 interface, suggesting that the tumor suppressor role of Fat and Dachsous may not require
their
348 direct interactions.”

First, in most situations Fat-Ds binding is thought to modulate signaling through the Fat or Ds ICDs. In that context, finding critical mutations in other parts of the Fat4 or Dchs1, as has been done for fly Fat, is not surprising, and does not rule out a critical role for binding. As for the Amotl1 and RET studies, were the effects ever shown to be independent of Dchs proteins? Better might be the fly studies, where simultaneous removal of both Fat and Ds has stronger effects on growth than removal of Ds. But even that result is complicated by the possibility of opposing activities of the Fat and Ds ICDs.

Second, please provide and state references here. Zhang et al for Fat4 and RET was not mentioned previously and is not in the reference list.

We thank the reviewer for spotting the omitted reference. We have added the reference in the correct place.

4) Null binding

“Through our mutational studies, we isolated and characterized null-binding
349 L379R and R297A mutants in Fat4 and Dchs1, respectively.”

The term “null-binding mutants” implies the complete absence of binding, while I think the authors can only claim a strong reduction, especially in the context of full-length proteins, where the sole analysis is reduced colocalization at cell interfaces.

We agree that this was strong wording given that the ITC experiments were done with the EC1-4 domains and not the full-length protein. We changed this to read: Through our mutational studies, we identified Fat4 L379R and Dchs1 R297A mutants that abrogated interactions between Fat4(EC1-4) and Dch1(EC1-4).

The reduced binding of mutant Fat4 EC1-4 to Dchs1 on cells in Extended Fig 5 is not very convincing, as the cells shown also have, in contrast to the controls, almost no Dchs1 on the cell surface.

We thank the reviewer for this assessment. We have updated the figure to depict another region from the same image file that displays cell surface staining comparable to the control.

Reviewer #2 (Remarks to the Author)

The authors have comprehensively responded to my comments, as well as those of the other reviewers. All my major and minor points have been adequately addressed. I have two *very*

minor points with the revisions to mention:

1. In response to reviews, the authors cut the resolution of one of their crystal structures, effectively removing data. There is a well-documented "perverse incentive" (PMID: 26209821) among scientists to cut their data, which usually results in decreases in Rmerge and in R/Rfree, to satisfy reviewers. High Rmerge and Rfree values in the highest resolution bin do not serve as a valid reason to remove valid data when CC1/2 values indicate otherwise. However, I respect the choice of the author's response to reviewer's comments.
2. Authors responded that updated validation reports are provided. I could not see these reports. Likely a problem with the manuscript submission system.

Reviewer #3 (Remarks to the Author)

The revised manuscript by Medina et al. entitled "Structure of the planar cell polarity cadherins Fat4 and Dachshous1", and the detailed rebuttal letter provided missing details. All issues raised in my report have been satisfactorily addressed and the manuscript has been improved. I would like to point out one minor thing in extended data table 6. Usually, deltaG values are also given in thermodynamic table along with deltaH and deltaS (although those can be easily calculated by deltaH and deltaS), and appropriate units should be provided with deltaH and deltaS, such as cal/mol and cal/mol K.

We updated Extended Data Table 6 to include the correct units and deltaG values.